# Particle filter based volcanic ash emission inversion applied to a hypothetical sub-Plinian Eyjafjallajökull eruption using the Ensemble for Stochastic Integration of Atmospheric Simulations (ESIAS-chem) version 1.0

Philipp Franke[1,2], Anne Caroline Lange[1,2], and Hendrik Elbern[1,2]

[1]Institute of Energy and Climate Research: Troposphere (IEK-8), Forschungszentrum Jülich GmbH, 52425 Jülich, Germany
[2]Rhenish Institute for Environmental Research at the University of Cologne, Aachener Str. 209, 50931 Köln, Germany

**Correspondence:** Philipp Franke (p.franke@fz-juelich.de)

**Abstract.** A particle filter based inversion system is presented, which enables to derive time- and altitude-resolved volcanic ash emission fluxes along with its uncertainty. The system assimilates observations of volcanic ash column mass loading as retrieved from geostationary satellites. It aims to estimate the temporally varying emission profile endowed with error margins. In addition, we analyze the dependency of our estimate on wind field characteristics, notably vertical shear, within variable observation intervals. Thus, the proposed system addresses the special challenge of analyzing the vertical profile of volcanic ash clouds given only 2D high temporal resolution column mass loading data as retrieved by geostationary satellites. The underlying method rests on a linear-combination of height-time emission finite elements of arbitrary resolution, each of which is assigned to a model run subject to ensemble-based space-time source inversion. Employing a modular concept, this setup builds the Ensemble for Stochastic Integration of Atmospheric Simulations (ESIAS-chem). It comprises a particle smoother in combination with a discrete-grid ensemble extension of the Nelder-Mead minimization method. The ensemble version of the EURopean Air pollution Dispersion - Inverse Model (EURAD-IM) is integrated into ESIAS-chem but can be replaced by other models. As initial validation of ESIAS-chem, the system is applied to simulated artificial observations of both ash-contaminated and ash-free atmospheric columns using identical twin experiments. Thus, in this idealized initial performance test the underlying meteorological uncertainty is neglected. The inversion system is applied to two notional sub-Plinian eruptions of the Eyjafjallajökull volcano, Iceland, with strong ash emission changes with time and injection heights. It demonstrates the ability of ESIAS-chem to retrieve the volcanic ash emission fluxes from the assimilation of column mass loading data only. However, the analyzed emission profiles strongly differ in their levels of accuracy depending of the strength of wind shear conditions. While the error is only 10 %–20 % for the estimated emission fluxes under strong wind conditions it increases up to 60 % under weak wind shear conditions. In case of increasing wind shear, the performance of the analysis may benefit from extending the assimilation window, in which new observations potentially contribute valuable information to the analysis system. For our test cases using an artificial volcanic eruption, we found an assimilation window length of 18 hours, i. e. 10 hours after the eruption terminated, to be sufficient for analyzing the extent and location of the artificial ash cloud. In the performed test cases, the analysis ensemble predicts the location of high volcanic ash column mass loading in the atmosphere

with a very high probability of >95 %. Additionally, the analysis ensemble is able to provide a vertically resolved probability map of high volcanic ash concentrations to a high accuracy for both, high and weak wind shear conditions.

## 1 Introduction

Emission profiles of volcanic eruptions depend on multiple parameters, such as crater size or exit velocity of the emitted mass. Further, they depend on atmospheric stability and wind profile at the volcano. Many of these parameters are unknown or difficult to measure exactly. This renders the estimation of emission profiles of volcanic eruptions challenging for chemistry transport models in the context of data assimilation and inverse modelling for source estimation. Therefore, special methods for assessing the strength and vertical distribution of volcanic emissions are necessary. As volcanic eruptions contain enormous amounts of harmful trace gases and particulate matter, a detailed knowledge not only about the spatial and temporal variations of the emissions and its strength is needed but also accurate information about the analysis error of the emissions and the evolving volcanic ash cloud is required.

Typically, explosive volcanic eruptions occur as sequences of emissions with highly varying ejection mass and height. Only limited observations of volcanic ash emission parameters are available (e.g. eruption plume heights retrieved from radar measurements, Arason et al., 2011), which are affected by their specific uncertainties and limitations, e. g. by orographic shielding in the case of radar observations. Thus, eruption models are applied to simulate volcanic emissions. These may be inferred from statistical data (e. g., Sparks et al., 1997; Mastin et al., 2009) or physical processes (e. g., Woodhouse et al., 2013; Folch et al., 2016). Statistical models base on observational data from historic volcanic eruptions, which are sparse and show a large variance in eruption rate for given plume heights. For example, Mastin et al. (2009) calculated an uncertainty by a factor of four in estimating the emission rate for a plume height of 25 km using their statistical model. Physical plume-scale models require orographic details of the volcano (e. g. crater size) but also meteorological fields and parameters (e. g. wind entrainment coefficients), which are often poorly known and render these models highly uncertain. Costa et al. (2016) identified the wind entrainment coefficient as main source of uncertainty leading to up to two orders of magnitude differences for the estimation of mass eruption rates for weak volcanic eruptions. In their analysis of the eruptions of the Eyjafjallajökull, Iceland, in 2010 and Grímsvötn, Iceland, in 2011, Woodhouse et al. (2015) found a comparable range of uncertainty depending on the choice of the wind entrainment coefficients.

Another potential way to constrain volcanic ash emissions is the use of observations of volcanic ash in the atmosphere. Advanced numerical analysis techniques for quantitative and stochastic estimation of volcanic ash concentrations and emissions use mostly satellite observations of column mass loading via data assimilation methods (e. g., Wilkins et al., 2016a). Column mass loading observations as available from, for example, the Spinning Enhanced Visual and InfraRed Imager (SEVIRI) on board Meteosat Second Generation (Schmetz et al., 2002) are beneficial for source inversions as they provide measurements of the horizontal extent of the volcanic ash cloud with a frequency as high as 15 minutes, which is used for analyzing the temporal evolution of the volcanic eruption column. However, in contrast to lidar observations, e. g. from the Cloud Aerosol LIdar with Orthogonal Polarization (CALIOP) instrument on board CALIPSO satellite (Winker et al., 2009) or the ground (e.

g. via the European Aerosol Research Lidar Network, EARLINET[1]), column mass loading observations rarely provide information about the vertical distribution of volcanic ash and are mostly limited to cloud top heights (e. g., L. J. Ventress, 2016; Piontek, 2021). Therefore, multiple data assimilation / source inversion methods make assumptions about the vertical extent of the volcanic ash cloud (e. g., Schmehl et al., 2012; Wilkins et al., 2016b; Zidikheri et al., 2016). In addition to column mass loading observations of volcanic ash clouds, Zidikheri et al. (2017a) suggested to use brightness temperature measurements to distinguish regions with high mass load from those with low mass load. All these observations are influenced by water cloud cover, which limits the detection of volcanic ash in the atmosphere.

First estimations of volcanic ash emissions from the 2010 Eyjafjallajökull eruption in a high temporal and vertical resolution were made by Stohl et al. (2011) and later by Kristiansen et al. (2012) and Kristiansen et al. (2015). Their algorithm bases on the inversion technique of Eckhardt et al. (2008), in which an optimal combination of distinct emission packages is estimated using a least squares method. The method showed to provide reliable a posteriori estimates of the time-varying emission profiles. Stohl et al. (2011) include errors from a priori estimates, retrieval errors and model errors and discussed results in terms of relative error reduction subject to assumptions made. Schmehl et al. (2012) initiate the volcanic ash analysis using an ensemble of simulations with random emission strengths and wind fields. Their best estimate of the volcanic ash concentration is found iteratively using a "genetic algorithm variational approach". Herein, rather strong assumptions on the emission profile are made: the emissions are kept fixed for the simulation duration; emissions are placed into a single model layer; wind fields are only adjusted in the model layer containing volcanic ash emissions. However, the method provides a quick and easy to implement first estimate of the volcanic ash concentrations in the atmosphere. Yet, the strong assumptions may render the approach unfeasible for longer lasting volcanic eruptions in which the emissions vary more strongly. Another data assimilation method for estimating the volcanic ash emissions was proposed by Lu et al. (2016). They developed an adjoint-free, ensemble-based four-dimensional variational data assimilation (4D-var) method. The method showed reliable estimates of the true emission profile in their experiments using synthetic, vertically integrated satellite observations. However, they do not address the uncertainty estimate of their analysis.

Zidikheri et al. (2016) and later Zidikheri et al. (2017b) developed an inversion system that aims to analyze the horizontal distribution of volcanic ash column mass loading rather than the emission strength. This study was extended by Zidikheri et al. (2017a) to additionally estimate the height and the particle size distribution of volcanic ash emissions using a parameter refinement method. Here, an ensemble of source parameter values has been applied. Using a proper metric (in their case the pattern correlation coefficient) the ensemble is evaluated against observations. The best fitted ensemble member is taken as analysis. The method is easy to implement for a fast analysis of a volcanic eruption as only the upper and lower bounds of the considered source parameters need to be defined. However, the number of model runs used to find the analysis increases exponentially with the number of parameters. Rough estimates of the parameters' uncertainty are provided by the spread of the top 2 % ensemble members with respect to the metric (Zidikheri et al., 2017b), which does not take uncertainties in the observed quantities into account.

Wilkins et al. (2014) used the "data insertion" method, in which observed volcanic ash column mass loadings act as virtual

---

[1]https://www.earlinet.org/index.php?id=earlinet_homepage

sources for volcanic ash with a predefined vertical distribution. The algorithm was successfully applied to the eruptions of Eyjafjallajökull, Iceland, 2010 (Wilkins et al., 2016b) and Grímsvötn, Iceland, 2011 (Wilkins et al., 2016c). Given the lack of vertical information in column mass loading retrievals of volcanic ash, the data insertion method needs assumptions about the vertical distribution of the volcanic ash content in the atmosphere. Thus, this larger source of uncertainty for the volcanic ash analysis is ignored. The data insertion scheme has also been implemented as a first step towards an ensemble-based data assimilation scheme in the FALL3D-8.0 atmospheric transport model (Prata et al., 2021).

Fu et al. (2017) developed a mask-state algorithm for ensemble Kalman Filters to reduce the size of the state vector to be optimized. More recent applications of the ensemble Kalman Filter and its variants are provided by Pardini et al. (2020) and Osores et al. (2020). By estimating the source parameters of the volcanic eruption, the approaches using the ensemble Kalman Filter assume constant emission parameters between two assimilation steps. This is a rather strong assumption on the emissions especially if observational data is sparse or far away from the volcano. However, keeping this assumption in mind the ensemble Kalman Filter methodology provides an estimate on the analysis uncertainty.

A general framework for calculating uncertainties of volcanic ash concentrations for constant volcanic ash emissions given "any model and any observational data" was proposed by Denlinger et al. (2012). Bursik et al. (2012) applied the "polynomial chaos quadrature weighted estimate" (PCQWE) method to volcanic ash emissions. This approach was further extended by Stefanescu et al. (2014) to take uncertainties in the wind fields into account as well. An extension of the polynomial chaos quadrature method was proposed by Madankan et al. (2014) to generate hazard maps of volcanic ash in the atmosphere. The PCQWE method aims to map uncertainties in the input parameters of a volcanic eruption onto the volcanic ash concentrations without accounting for constraining volcanic ash observations. Additionally, Dare et al. (2016) investigated the influence of meteorological ensemble forecasts on the dispersion of volcanic ash. They found that not only the ensemble statistics should be evaluated but also the single ensemble members, which may contribute significant information to the distribution of volcanic ash.

Although these studies applied highly advanced data assimilation and source inversion methods for analyzing the emission strength and the uncertainty of volcanic ash dispersion forecasts, a joint assessment of both, emission strength and its uncertainty, in a high temporal and vertical resolution has not yet been evaluated. Thus, this contribution aims to fill this gap by providing such estimates of emission strength and its uncertainty in a high temporal and spatial resolution resulting in height-resolved probability maps of volcanic ash concentrations.

Section 2 describes the full stochastic inversion system ESIAS-chem and the methods applied. The potential and limitations of ESIAS-chem are shown by identical twin experiments in section 3. A discussion of the results and conclusions will be given in section 4.

## 2   Methodology

The Ensemble for Stochastic Integration of Atmospheric Simulations (ESIAS) is designed to control simultaneous and inter-active runs of ultra-large ensembles of complex atmospheric models. ESIAS comprises a meteorological (ESIAS-met) and an

**Table 1.** Nomenclature for variables used in this study.

| | |
|---|---|
| $N_T$ | number of time steps |
| $K_{max}$ | number of vertical model layers |
| $N_{emis}$ | number of pairwise distinct emission packages ($N_{Emis} = N_T * K_{max}$) |
| $\mathbf{s}$ | vector of pairwise distinct emission packages of default strength |
| $\mathbf{a}$ | vector of scaling factors for the emission packages with components $a_i$ ($\hat{\mathbf{a}}$: its optimal values) |
| $H$ | observation operator |
| $M$ | operator for the source-receptor model |
| $\mathbf{R}$ | observation error covariance matrix |
| $\mathbf{y}$ | observation vector |
| $c$ | default emitted mass of ash (constant) |
| $\mathbf{e}_i$ | $i^{th}$ unit vector |
| $\mathbf{x}$ | modelled volcanic ash concentration |
| $x_k$ | modelled volcanic ash concentration in model layer $k$ |
| $\hat{x}$ | column mass loading of the model state |
| $\Delta z_k$ | thickness of layer $k$ in meters |
| $\mathbb{N}_0^N$ | bulk of positive integers (including 0) |
| $\mathbb{R}^N$ | bulk of real numbers |
| $p(\cdot)$ | probability density function |
| $N_{Ens}$ | size of the analysis ensemble (i. e. number of ensemble members) |
| $(\cdot)^{(i)}$ | superscript indicating ensemble member $i$ |
| $\delta(\cdot)$ | Kronecker delta function |
| $w^{(i)}$ | weight of ensemble member i according to the particle filter formulation |
| $t_i$ | time step $i$ |
| $\mathbf{B}$ | estimated covariance matrix for the scaling factors $\mathbf{a}$ |
| $\sigma_{y_i}$ | observation error for observation $y_i$ (diagonal elements of $\mathbf{R}$) |

atmospheric chemical part (ESIAS-chem). One main feature of ESIAS is the potential to include data assimilation and source
inversion modules. The emphasis of this paper is placed on ESIAS-chem for probabilistic atmospheric chemistry-transport-
diffusion simulations with data assimilation. In the following, the main components of the ESIAS-chem system are introduced,
which include an ensemble of emission packages, a modified version of the Nelder-Mead minimization algorithm, and a par-
ticle filter and resampling algorithm. Further, the full stochastic particle smoother algorithm of ESIAS-chem for probabilistic
volcanic ash analyses is described. Finally, the metrics for analyzing the systems performance is summarized. The reader is
referred to Tab. 1 for a summary of the used variables.

## 2.1 Model description

### 2.1.1 Ensemble of emission packages

ESIAS-chem is initiated by simulating the dispersion of normalized emissions from a set of $N_{emis}$ pairwise distinct emission packages (emission scenarios in the terminology of Stohl et al., 2011) each of which with a default mass of ash. More specific, the eruption plume is discretized into $N_T$ time steps and $K_{max}$ vertical layers. The simulation is realized by an ensemble, in which each ensemble member simulates the dispersion of one single emission package. Thus, each emission package covers a unique time and height spot in the emission profile. We refer to this ensemble as ensemble of emission packages. This approach accounts for multiple maxima in the vertical distribution of volcanic ash emissions. These may occur within discretized model time steps if the emission strength varies quickly in strength and plume height. Each member of the ensemble of emission packages simulates the dispersion of volcanic ash concentrations released by a single emission package. A similar approach for estimating the volcanic ash column mass loading was used by Stohl et al. (2011) and Kristiansen et al. (2015) aiming to estimate the optimal emission profile. Contrary to their analysis, the focus of ESIAS-chem lays on the predictability of volcanic ash emissions and the resulting volcanic ash concentrations.

In order to find the optimal emission profile, the cost function $J(\mathbf{a})$

$$
\begin{aligned}
\hat{\mathbf{a}} = \operatorname{argmin}\left(J(\mathbf{a})\right) &= \operatorname{argmin}\left((H\mathbf{x}(\mathbf{a}) - \mathbf{y})^T \mathbf{R}^{-1}(H\mathbf{x}(\mathbf{a})) - \mathbf{y})\right) \\
&= \operatorname{argmin}\left(\left(\sum_{i=1}^{N_{emis}} HM[a_i\mathbf{s}_i] - \mathbf{y}\right)^T \mathbf{R}^{-1}\left(\sum_{i=1}^{N_{emis}} HM[a_i\mathbf{s}_i] - \mathbf{y}\right)\right),
\end{aligned}
\tag{1}
$$

is to be minimized, with the source-receptor model $M$ mapping the unique emissions $\mathbf{s}_i$ of member $i$ of the ensemble of emission packages onto the model state. Herein, $\mathbf{s}_i = (0, \ldots, 0, c, 0, \ldots, 0)^T = c * \mathbf{e}_i$ is a $K_{max} * N_T = N_{emis}$ dimensional vector with $c = const$ the default mass of ash of each emission package and $\mathbf{e}_i$ the $i^{th}$ unit vector. Scalar $a_i$ is the scaling factor for the $i^{th}$ emission package. For the optimization, $a_i$ values are packed into vector $\mathbf{a}$ resulting in optimal values $\hat{\mathbf{a}}$. Further, $\mathbf{R}$ is the observation error covariance matrix and $H$ denotes the observation operator for volcanic ash column mass loading, which is given by

$$
\hat{x} = H\mathbf{x}(\mathbf{a}) = 10^6 \sum_{k=1}^{K_{max}} x_k \Delta z_k,
\tag{2}
$$

where $x_k$ is the modeled concentration of volcanic ash in $[\mu\text{g m}^{-3}]$ and $\Delta z_k$ is the thickness of model layer $k$ in $[\text{m}]$. Please note that the vertical and temporal resolution of the emissions can be varied by changing the parameters $K_{max}$ and $N_T$, thus, making them independent from the vertical and temporal resolution of the model. Matrix $\mathbf{R}$ accounts for the impact of retrieval errors of the volcanic ash column mass loading and is considered diagonal. It can be made spatially and temporally dependent, to account for assumed increased retrieval errors due to water cloud influences, particularly thick umbrella ash clouds above or in the vicinity of the volcano or interference of other aerosols or mineral dust. In our study, we have made assumptions about the observation error (including retrieval error). In applications to real volcanic eruptions, the use of retrieval errors provided

by the observations is highly encouraged. Starting from a scalar column load value as exclusive data source, we considered estimation uncertainties of the derived height profile presented here as an order of magnitude larger than retrieval errors in this idealized experiments, especially if the number $K_{max}$ determines some multiple of O(10) layers. The observation error can also be incorporated in constructing the ensemble, as in general any ensemble data assimilation procedure can straightforwardly account for the retrieval uncertainty by artificially perturbing retrievals of column mass loading, where the random perturbation is scaled by the assumed statistics of retrieval errors. Clearly, this must not be the only means to generate the ensemble, as this accounts only for a fraction of the overall uncertainty, resulting in underdispersive ensembles. Although the algorithm is designed for column mass loading observations, it is as well applicable to observations of any volcanic ash related quantity, e. g. brightness temperature as proposed by Zidikheri et al. (2017a).

### 2.1.2 Nelder-Mead algorithm

The minimization problem posed by (1) is quadratic within the limits of being bounded due to positive semi-definiteness of all components. Quasi-Newton methods, including a bounded variant proved less efficient, as a background state reasonably close to the "truth" for a tangent-linear approximation to hold, is typically unknown. This missing a priori knowledge cannot serve any preconditioning requirements other than highly speculative inferences from assumed eruption type and strengths scenarios. With an increasing number of model levels with their (positive semi-definite) concentrations to be attributed, while column values as given data are single scalars only, the ill-conditioning of the minimization problem increases drastically and a much needed reasonable background information prior to the volcanic eruption is hardly available. Also simple smoothness assumptions of the vertical profile are often invalid for ash clouds, at least during early stages. As minimization tests with the Nelder-Mead method performed clearly best, without getting lost in drastically elongated minima as introduced by underdetermined degrees of freedom through vertical level concentrations, the algorithm by Nelder and Mead (1965) was applied to the inversion problem. The Nelder-Mead minimization algorithm is a combinatorial optimization method without constraints and without the need to compute the function derivatives. It has proven to be robust, especially in cases where the function to be minimized has discontinuities or the function values are noisy (see McKinnon, 1998). This is expected to be likely in highly variable volcanic eruptions especially given highly uncertain, and thus noisy, observations. Additionally, the Nelder-Mead algorithm can easily account for bounded regions, in our case positive semi-definite ash loads, and needs relatively few function evaluations (mostly 1-2 function evaluations per iteration, Lagarias et al., 1998).

The idea of the algorithm is to move a simplex on the surface of the cost function to find an improved model state in a $N_{emis}$-dimensional space. The version of the Nelder-Mead method used in this study follows Gao and Han (2012) and utilizes adaptive parameters controlling the step size for each iteration of the minimization. The version has been implemented for parallel operation (Klein and Neira, 2014; Lee and Wiswall, 2007). In our application, the Nelder-Mead algorithm is used to find the optimal combination of the pairwise distinct emission packages. This is accomplished by assigning a factor $a_i$, which needs to be scaled by the algorithm, to each emission package.

Due to its simplicity, the Nelder-Mead algorithm is easy to implement but it is likely to find a local rather than the global minimum of the cost function (which is also a problem for least-square minimization techniques with poor initial guesses, as

for volcanic eruptions). Thus, we have added some adjustments to the algorithm. First, we perform the minimization only for integers (including 0). Thus, only integer values are accepted for the scaling factors $a_i$ of the emission packages. By applying this constraint it is assumed that the introduced errors are of lower order than the error introduced by the temporal and vertical resolution of the emission packages. Further, the minimization is restarted with larger perturbations of the vertices (edges of the simplex) if the algorithm is trapped in a local minima. Finally, the minimization is started for an ensemble of Nelder-Mead analysis. As perturbed observations are used as input to the minimization procedure, the solutions (here emission profiles) produced by the analysis ensemble are assumed to map the uncertainty given by the observations onto the emission rates (cf. Sect. 2.1.4). Thus, the minimization algorithm is called hereafter discrete-grid ensemble Nelder-Mead method (DENM).

### 2.1.3  Particle filter

The particle filter methodology, also known as sequential importance resampling, is used as a non-Gaussian data assimilation method for large ensemble simulations of the atmospheric state. Please note that the term ensemble in this section defines a full model simulation and does not refer to the ensemble of emission packages defined in the Sect. 2.1.1. The particle filter method was proposed by Gordon et al. (1993) and further popularized in oceanography and meteorology by van Leeuwen (2009). It develops from Bayes' Theorem

$$p(\mathbf{x}|\mathbf{y}) = \frac{p(\mathbf{y}|\mathbf{x})p(\mathbf{x})}{\int p(\mathbf{y}|\mathbf{x})p(\mathbf{x})d\mathbf{x}}, \tag{3}$$

where p($\cdot$) denotes the probability density function (PDF), $\mathbf{y}$ the observations, and $\mathbf{x}$ the model state. The a priori PDF is approximated by an ensemble of $N_{ens}$ model runs

$$p(\mathbf{x}) = \frac{1}{N_{ens}} \sum_{i=1}^{N_{ens}} \delta(\mathbf{x} - \mathbf{x}^{(i)}), \tag{4}$$

where $\delta(\cdot)$ denotes the Kronecker delta function and $\mathbf{x}^{(i)}$ is the model state of particle (ensemble member) $i$.
Applying (4) results in

$$p(\mathbf{x}|\mathbf{y}) = \sum_{i=1}^{N_{ens}} \frac{p(\mathbf{y}|\mathbf{x}^{(i)})}{\sum_{j=1}^{N_{ens}} p(\mathbf{y}|\mathbf{x}^{(j)})} \delta(\mathbf{x} - \mathbf{x}^{(i)}) = \sum_{i=1}^{N_{ens}} w^{(i)} \delta(\mathbf{x} - \mathbf{x}^{(i)}), \tag{5}$$

where an individual weight $w^{(i)}$ is applied to each ensemble member. Thus, each ensemble member is weighted by the normalized likelihood of its current model state. The ensemble statistics can now be computed using the ensemble member weights. For example, the ensemble mean is

$$\overline{\mathbf{x}} = \sum_{i=1}^{N_{ens}} w^{(i)} \mathbf{x}^{(i)}.$$

It is noted that in the particle filter method no assumptions of the statistical forecast error characteristics and the observation error were made (the errors do not need to be normally distributed and the model state does not need to be unbiased as other data assimilation methods require). Further, the particle filter method is directly applicable to nonlinear models.

In particle filters, filter degeneracy often occurs (cf. Bengtsson et al., 2008; Snyder et al., 2008; Bickel et al., 2008), especially in high dimensional problems. Several methods exist to reduce filter degeneracy (see e.g. van Leeuwen, 2009, for a review) and the reader is referred to the original papers for more information. In ESIAS-chem, the particle filtering and resampling steps are applied after the ensemble of optimal emission profiles has been found by the DENM algorithm. A weight $w^{(i)}$ is assigned to each optimal emission profile. Residual resampling (Liu and Chen, 1998) is used to replace emission profiles leading to small weights by emission profiles with high weights (this step includes perturbing duplicated emission profiles). After resampling, the weights are normalized again ($w^{(i)} = 1/N_{Ens}$). Thus, the statistical informative value of the analysis ensemble is preserved. Qualitatively, the strategy of particle filtering applied here can be expressed as follows: By replacing the valueless ensemble members of the analysis (i. e. those with too little weight) each ensemble member has comparable skill to match the observations. Hence, the probability of an event (e. g. volcanic ash concentrations above a certain threshold) can directly be extracted from the relative number of ensemble members that simulate this event.

### 2.1.4 ESIAS-chem

ESIAS-chem is designed as a flexible analysis system for quantitative volcanic ash assessments, along with an uncertainty quantification of the analyzed emission flux profile. ESIAS-chem is constructed such that it is applicable to other scenarios of accidentally released matter and constituents, given constraining observations are available. Further, it is capable to be coupled with ensembles of meteorological fields to account for additional uncertainties resulting from meteorological forecasts. However, this idealized investigation focuses on the ability of the system to reconstruct the emission profile and its uncertainty under perfect meteorological conditions. Thus, no meteorological ensemble is used at this stage.

The aerosol dynamics (nucleation, accumulation, deliquescence) and aerosol chemistry in EURAD-IM (EURopean Air pollution Dispersion - Inverse Model), which poses the model core of ESIAS-chem, is based on MADE (Modal Aerosol Dynamics model for Europe, Ackermann et al., 1998, with substantial update of the thermodynamical part by Friese and Ebel, 2010), has been switched off for two reasons: numerical efficiency in an ensemble context and specifics of volcanic ash properties cannot be expected to be reasonably well featured by a general pollutant aerosol module like MADE. Ideally, a full volcanic ash aerosol dynamics and chemistry model as proposed by Schmidt (2013) would be in place, along with its not existing adjoint. Yet we consider the error to be negligible within the evolution time frame addressed in our idealized study.

The workflow of the system is illustrated in Fig. 1. Once a volcanic eruption is detected ($t = t_0$), the system is started by generating the ensemble of emission packages (cf. section 2.1.1) of default mass of volcanic ash. As long as no observations are available, this ensemble of emission packages serves as an estimation predictor of the maximum possible volcanic ash extent without providing quantitative volcanic ash estimates. Once new observations become available, the system is restarted at time $t = t_0$. The previously calculated ensemble of emission packages is reused, now integrated forward in time until the observation time ($t = t_{i+1}$). Further emission packages (i. e. members of the ensemble of emission packages) are included to account for the latest emissions in the interval $[t_i, t_{i+1}]$.

The ensemble of emission packages is compared with the observations to calculate (1), which is to be minimized using the DENM method. As this algorithm is optimized for the estimation of column integrated ash loading in a considerably underde-

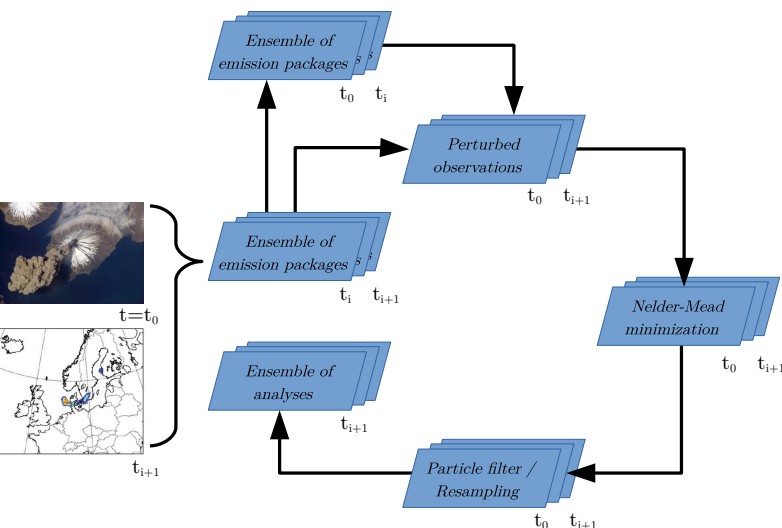

**Figure 1.** Schematic of the ESIAS-chem analysis workflow. The analysis is initiated at time $t = t_0$ and restarted, when new observations become available (left side, cf. Sect. 2.1.1). Here, $t_{i+1}$ corresponds to the observation time. Previously calculated simulations with emission packages within the time interval $t_0 - t_i$ may be restored (upper panel). Simulated volcanic ash is compared with perturbed observations for the whole simulation (i. e. from $t_0$ to $t_{i+1}$; upper center panel). The resulting volcanic ash concentrations are passed to the DENM minimization algorithm that produces an ensemble of emission profile analyses (right panel, cf. Sect. 2.1.2) by finding an optimal combination of the pairwise distinct emission packages. This ensemble of emission profile analyses is evaluated by the particle filter and resampling method to assign a weight to each emission profile according to the fit of the resulting volcanic ash to the observations. Emission profiles are replaced if their corresponding volcanic ash content does not fit well to the observations (lower panel, cf. Sect. 2.1.3). Credits for volcano image: NASA (https://www.nasa.gov/multimedia/imagegallery/image_feature_756.html; Cleveland volcano, Aleutian Islands, latest access: 13 January, 2021).

termined control system, a regularization term is added to the cost function (1), leading to

$$J(\mathbf{a}) = \mathbf{a}^T \mathbf{B}^{-1} \mathbf{a} + \left( \sum_{i=1}^{N_{emis}} HM[a_i \mathbf{s}_i] - \mathbf{y} \right)^T \mathbf{R}^{-1} \left( \sum_{i=1}^{N_{emis}} HM[a_i \mathbf{s}_i] - \mathbf{y} \right). \tag{6}$$

This choice restricts the scaling factors $\mathbf{a}$ to vary too strongly. In first tests without the regularization term, the emission rates have partly increased to unrealistic high values. Therefore, the $\mathbf{B}$-matrix was chosen in a sequence of sensitivity tests, in which the influence of the regularization term on the emission profile was evaluated. Best results have been found by choosing $\mathbf{B}$ as diagonal matrix $\mathbf{B} = diag(10)$. Please note that the chosen diagonal form of the $\mathbf{B}$-matrix led to reasonable results for the artificial emission profile used in this study. However, for realistic applications a more elaborated evaluation of a properly

chosen **B**-matrix is required and straightforwardly applicable. In this performance test, the only purpose of the matrix serves to restrict the scaling factors **a** not to vary too strongly. In addition, the regularization term was chosen in order to maintain a
265 suitable spread of the analysis ensemble.

The minimization is initialized with a set of arbitrarily varying scaling factors **a** for the pairwise distinct emission packages. The algorithm was tested using a time-varying initial emission profile with umbrella-shaped vertical mass distribution. Due to the chosen true emission profile in this idealized study (cf. Sect. 3), the minimization using the initial emission profile with umbrella-shaped vertical mass distribution shows larger errors. In the application of the algorithm to a real volcanic
eruption, the performance of the analysis using an umbrella-shaped initial emission profile may exceed the performance using an arbitrary emission profile. Hence, ESIAS-chem is designed to adjust the initial emission profile to the characteristics of the current volcanic eruption. In addition, the observation errors are represented by perturbed observations (cf. Houtekamer and Mitchell, 1998), which are assimilated by ESIAS-chem leading to larger ensemble spreads.

Once an improved emission profile has been found by the DENM minimization, a particle filter step is applied to the analysis
ensemble. The weights, which result from the filtering step, are applied to the analyzed emission profiles. If new observations become available the assimilation window may be elongated to use the new observations for updating the emissions within the whole assimilation window $[t_0, t_{i+1}]$.

## 2.2 Metrics used for analyzing ESIAS-chem's performance

As ESIAS-chem is tested by identical twin experiments (cf. Section 3 for more details), results of the analysis are compared
to a "nature run". In this experimental setting, the nature run is considered to represent the truth. Synthetic observations are simulated by extracting volcanic ash column mass loading data from this nature run. These synthetic observations show only a small fraction of the data that is used to validate the analysis ensemble. Thus, the following test procedures compare volcanic ash simulated by the analysis ensemble with volcanic ash simulated by the nature run rather than only with the extracted observations.

The results of the stochastic inversion method are validated using different measures on the ensemble mean. The pattern correlation coefficient (pcc, cf. Zidikheri et al., 2016) provide information about the accuracy of the horizontal extent of the volcanic ash cloud. The pcc is defined by (Zidikheri et al. (2016))

$$pcc = \frac{< \mathbf{va}'_{\overline{x}}, \mathbf{va}'_y >}{|\mathbf{va}'_{\overline{x}}||\mathbf{va}'_y|}, \tag{7}$$

with $\mathbf{va}' = \mathbf{va} - \overline{\mathbf{va}}$. Herein, the entries of the binary volcanic ash detection vector **va** for the ensemble mean (subscript $\overline{x}$) and
290 observations ($y$) are equal to 1 if the grid column contains volcanic ash above $0.2 \,\mathrm{g\,m^{-2}}$, which is the detection limit of volcanic ash column mass loading observations (Prata and Prata, 2012), and 0 otherwise. The averaged volcanic ash detection $\overline{\mathbf{va}}$ is calculated by

$$\overline{\mathbf{va}} = < \mathbf{1}, \mathbf{va} > / < \mathbf{1}, \mathbf{1} >, \tag{8}$$

where $\mathbf{1}$ denotes the vector with 1 on all entries and $< \cdot, \cdot >$ indicates the scalar product. The pattern correlation coefficient gives information about the compliance between the simulated and observed horizontal distribution of volcanic ash in the atmosphere. If the volcanic ash clouds, indicated by the column mass loading, of the nature run and the ensemble mean perfectly coincide, the pattern correlation coefficient equals 1. If ash cloud covers of the nature run and of the ensemble mean match nowhere, the pattern correlation coefficient equals 0.

The inner-cloud distribution of volcanic ash of the analysis ensemble mean is analyzed using the relative mean absolute error (RMAE). The RMAE is defined by

$$RMAE = 100 \frac{1}{N_y} \sum_{j=1}^{N_y} \left| \frac{\overline{x}_j - y_j}{y_j} \right|, \tag{9}$$

where $N_y$ is the number of grid columns in which volcanic ash column mass loading of the nature run $y_j \geq 0.2 \mathrm{\,g\,m^{-2}}$ and $\overline{x}_j$ is the column mass load of the analysis ensemble mean in grid cell $j$. The RMAE is as well calculated for volcanic ash concentrations, for which this threshold is $\geq 10 \mathrm{\,\mu g\,m^{-3}}$. In this case, $y_j$ corresponds to the volcanic ash concentration simulated by the nature run. The relatively low threshold to calculate the RMAE was chosen in order to increase the number of grid cells to be analyzed and to investigate the full volcanic ash cloud rather than only the area of high concentrations. The RMAE measures the relative difference of column ash mass loads (or volcanic ash concentrations) between nature run and the analysis ensemble mean, averaged over all grid cells. Higher values of the RMAE are a result of different height–time–mass emission patterns between the nature run and the analysis ensemble mean, given the assumed perfect meteorology used in this study.

The probability estimate of the ensemble analysis is investigated using the Brier score

$$BS = \frac{1}{N_y} \sum_{j=1}^{r} \sum_{i=1}^{N_y} \left( p_{j,i} - E_{j,i} \right)^2, \tag{10}$$

where $r$ is the number of verification classes, $p_{j,i}$ is the forecast probability of the ensemble for class $j$ to predict event $i$, and $E_{j,i}$ is the respective observed probability. The probability of the analysis ensemble to model volcanic ash concentrations within eight verification classes is analyzed. These classes are [10 $\mathrm{\mu g\,m^{-3}}$, 50 $\mathrm{\mu g\,m^{-3}}$], [50 $\mathrm{\mu g\,m^{-3}}$, 100 $\mathrm{\mu g\,m^{-3}}$], [100 $\mathrm{\mu g\,m^{-3}}$, 250 $\mathrm{\mu g\,m^{-3}}$], [250 $\mathrm{\mu g\,m^{-3}}$, 500 $\mathrm{\mu g\,m^{-3}}$], [500 $\mathrm{\mu g\,m^{-3}}$, 1000 $\mathrm{\mu g\,m^{-3}}$], [1000 $\mathrm{\mu g\,m^{-3}}$, 1500 $\mathrm{\mu g\,m^{-3}}$], [1500 $\mathrm{\mu g\,m^{-3}}$, 2000 $\mathrm{\mu g\,m^{-3}}$], and [2000 $\mathrm{\mu g\,m^{-3}}$, $\infty$). The observed probability is $E_{j,i} = 1$ if the nature run volcanic ash concentration is within a certain class and $E_{j,i} = 0$ otherwise. A perfect probabilistic forecast result in a Brier score close to 0. The small threshold values are chosen to see the performance of the ensemble for analyzing the full volcanic ash cloud. Further, the number of grid cells with large volcanic ash concentrations is limited, which renders the Brier score inapplicable. Finally, the forecast probability of the analysis ensemble is analyzed. The forecast probability is computed as the relative number of ensemble members predicting the event (i. e. the number of ensemble members forecasting volcanic ash concentrations within a certain class).

## 3 Validation of ESIAS-chem

The ability of ESIAS-chem to provide quantitative estimates of the volcanic ash emission uncertainty is explored by identical twin experiments. Identical twin experiments are necessary, yet not sufficient standard test procedures for validating spatio-temporal data assimilation and inverse modelling set-ups. They are idealized experiments as they rest on the "perfect model assumption" and its analog for the data side: exactly known accuracy and representativity. This provides a total knowledge of the "synthetic truths" as given by simulations with the same model and extraction of artificial "measurements/soundings" thereof. The term identical twin refers to the fact, that observations and a priori knowledge are constructed from the same model and input data, in which only the parameters to be optimized (emission profile in our case) differ. Given the identical twin assumption the experiment is then to be made realistic in all other respects. Daley (1991) concludes however, that identical twin experiments "err on the optimistic side" (loc. cit.). Yet, the applicability of ESIAS-chem to real volcanic eruptions will be shown in a companion paper.

### 3.1 Experimental setup

In this study, ESIAS-chem is online coupled to the regional air quality model EURAD-IM (EURopean Air pollution Dispersion - Inverse Model, Elbern et al., 2007). As we consider the differences of feedbacks of the ash clouds on the meteorological evolution as not critical on the forecast time scale in our idealized tests, the EURAD-IM is offline coupled with the Weather Research and Forecasting (WRF) model version 3.7 (Skamarock et al., 2008). Meteorological boundary conditions are taken from the ECMWF (European Centre for Medium-Range Weather Forecasts) analysis. The simulations have been run on the JUQUEEN supercomputer (Jülich Supercomputing Centre, 2015).

The ESIAS-chem system is tested for notional volcanic eruptions of the Eyjafjallajökull volcano, Iceland, on 15 April and 29 April 2010. These dates were chosen as different meteorological conditions with strong and weak winds at the volcano occurred. Column mass loading of volcanic ash in $[\mathrm{g\,m^{-2}}]$ is extracted as fictional observation data $y_i$ every 6 hours from a 'nature' run, simulated by the forward model of EURAD-IM. These data mimic cloud free retrievals from the Spinning Enhanced Visible and Infrared Imager (SEVIRI) onboard the geostationary Meteosat Second Generation satellite, with significantly larger time steps (six hours in this analysis), however. The synthetic observations from the nature run are extracted from the full domain including grid cells containing volcanic ash and those without volcanic ash. The information of ash free areas is necessary in order to avoid estimates of spurious emissions at false times and heights in the analysis. Quasi-continuous data streams of emissions and observations, as available in real applications, prohibit any attribution of time-height ash emissions to later ash column retrievals for test purposes. In fact, while the restriction of the data flux to 6 hourly time intervals in our test scenarios reduces the information used, it helps to attribute column mass loading observations to older and recent emissions within the chosen resolution. We found a six hours interval for column mass loading data supply practicable. For later operational purposes the use of the full high frequency data supply is readily adaptable.

The uncertainty of volcanic ash column mass loading observations is about 40 % (Western et al., 2015; Clarisse and Prata, 2016) or even higher (Wen and Rose, 1994; Kylling et al., 2014). For the identical twin experiments in this study, the obser-

vation error needs a special treatment because a relative error overemphasizes the system to low observed values. Further, no relative error for observations of no-volcanic ash observations can be obtained. Therefore, the following expression for the observation error $\sigma_{y_i}$ with a minimum error of $0.1\,\mathrm{g\,m^{-2}}$ is used

$$360 \quad \sigma_{y_i} = \max\left[\frac{(y_i * 0.4)^2}{\max_j(y_j * 0.4)}, 0.1\right]. \tag{11}$$

With this choice of observation error, the impact of low volcanic ash values observed at the edge of the volcanic ash cloud on the analysis is diminished. For applications to real volcanic eruptions, the observation error provided by the satellite retrieval per pixel should be considered.

The Hovmoeller-like plot of the nature run emission profile is shown in Fig. 2. It shows the variable emission rate by a height-time graphic above the volcano. The selected sub-Plinian eruption type (Bardintzeff and McBirney, 2000) is characterized by two short explosive phases between 02-04 UTC and 06-08 UTC reaching a height of approx. $8\,\mathrm{km}$ above the volcano.

The length of the assimilation window influences the performance of the inversion algorithm due to differences in vertical and horizontal mixing and vertical wind shear. Hence, the performance of ESIAS-chem is tested for different assimilation window lengths. All assimilation windows start at 00 UTC for the specific day and last for 6-36 hours. Fig. 3 illustrates the different assimilation window lengths, which differ in length by multiples of 6 hours. With increased residence time in the atmosphere, the volcanic ash at different heights becomes more horizontally split by wind shear. This effect can be exploited by increasing the assimilation window length. Contrary, vertical and horizontal mixing of volcanic ash may limit the benefit that is gained by increasing the assimilation window length. For example, if volcanic ash emitted by two different emission packages is mixed, it is impossible to attribute the volcanic ash to one or the other emission package. Once the volcanic ash emissions are

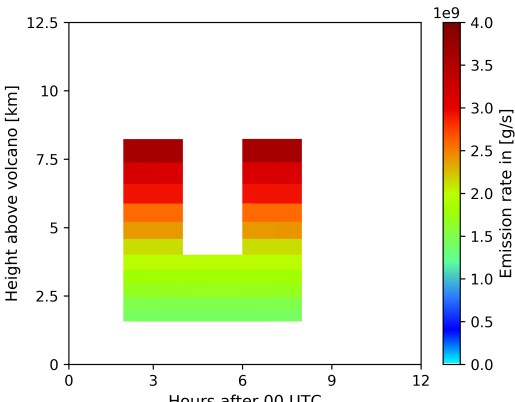

**Figure 2.** Hovmoeller-like plot of the volcanic ash emission profile used in the nature run to generate the synthetic observations simulating SEVIRI-like column mass loading data for the ESIAS-chem system tests. Shown is the emission rate (colored) for a given time (x-axis) and height above the volcano (y-axis).

optimized, a forecast is appended until 36 hours after the simulation start (hatched areas in Fig. 3). Thus, beneficial impacts of the inversion results for the analyses with differing assimilation window lengths can be assessed.

The first real weather test day to which ESIAS-chem is applied is 15 April 2010, which was characterized by strong west–north–westerly winds in Iceland. This is illustrated by the wind profile at the volcano (Fig. 4a) for the whole simulation length

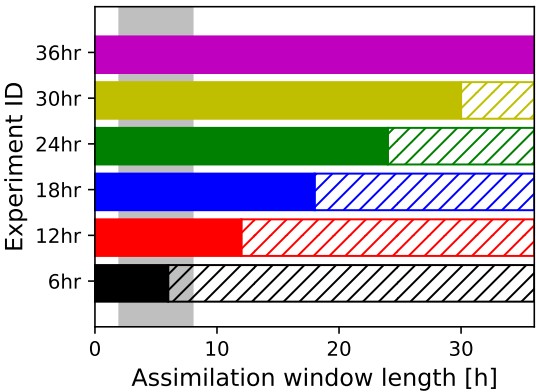

**Figure 3.** Illustration of the varying assimilation window lengths for the identical twin experiment (filled bars). After each assimilation window, a free forecast until 36 hours after the start of the simulation is appended (hatched bars). Color codes and experiment IDs correspond to the annotations in the subsequent figures. The gray background area indicates the duration of the volcanic eruption from hours 2 to 8.

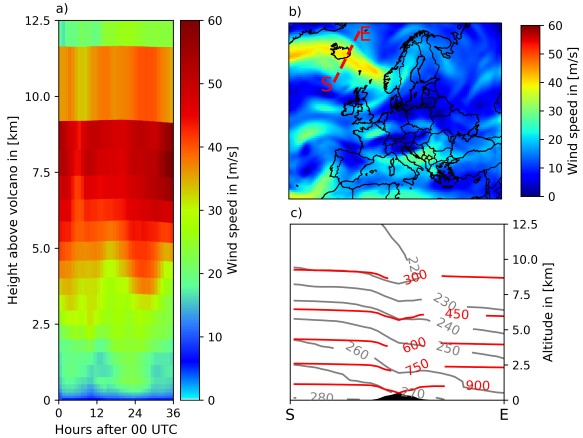

**Figure 4.** Meteorological conditions on 15 April 2010. (a) Wind speed above the volcano for the whole simulation period. (b) Wind speed at 500 hPa on 15 April 2010, 12 UTC, which corresponds to approx. 5 km above the volcano. (c) Vertical cross-section of isobars in [hPa] (red) and isotherms in [K] (grey) along the red line in b) on 15 April 2010, 12 UTC.

of 36 hours and the wind field over Europe at 500 hPa on 15 April 2010, 12 UTC (Fig. 4b). During this day, the polar front
and the polar jet stream are located above Iceland, driving the volcanic ash to travel fast southeast towards continental Europe,
with wind speeds of up to 60 m s$^{-1}$ at heights of 5–8 km above the volcano. Fig. 4c shows a vertical cross section of pressure
and temperature along the red line in Fig. 4b at 12 UTC on 15 April 2010. As indicated by the intersection of the isobars
and isotherms, 15 April 2010 is characterized by substantial vertical wind shear above Iceland, which is expected to ease the
distinction of volcanic ash emitted at different heights as seen from above.

In addition to the synoptic scenario on 15 April 2010, a second analysis of ESIAS-chem's performance is made for a hy-
pothetical sub-Plinian eruption of the Eyjafjallajökull volcano on 29 April 2010 (Fig. 5). A similar emission profile is taken
as depicted in Fig. 2, yet with slightly different emission rates. This day is characterized by weak winds of approximately
10 m s$^{-1}$ in the vicinity of the volcano, which is illustrated by Fig. 5. Thus, the emitted volcanic ash is only slowly transported.
Additionally, the vertical wind shear on 29 April 2010, 12 UTC, is low, as indicated by a higher barotropicity above Iceland
(Fig. 5c). The two dates were chosen because of their different wind patterns and the real eruption of the Eyjafjallajökull
that occurred during these days. Hence, the identical twin experiment provides an optimal case scenario for the application of
ESIAS-chem to real volcanic eruptions.

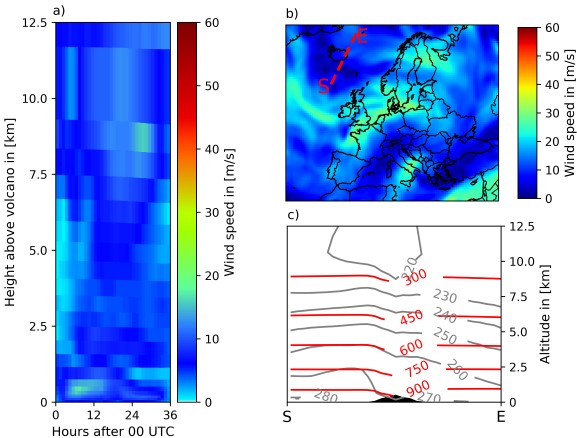

**Figure 5.** Meteorological conditions on 29 April 2010. (a) Wind speed above the volcano for the whole simulation period. (b) Wind speed
at 500 hPa on 29 April 2010, 12 UTC, which corresponds to approx. 5 km above the volcano. (c) Vertical cross-section of isobars in [hPa]
(red) and isotherms in [K] (grey) along the red line in b) on 29 April 2010, 12 UTC.

## 3.2 Evaluation of ESIAS-chem

### 3.2.1 Volcanic ash dispersion

The ability of the analysis ensemble mean to predict the volcanic ash dispersion is investigated using the pattern correlation coefficient (pcc) and the relative mean absolute error (RMAE) introduced in Section 2.2. The pattern correlation coefficient is shown in Fig. 6 for the two analysis days. The lines in Fig. 6 indicate results for different assimilation window lengths as illustrated by Fig. 3. Fig. 6 shows a constantly large pattern correlation coefficient $> 0.95$ after the artificial eruption terminated at 08 UTC for both analysis days, except for assimilation window lengths of 6 and 12 hours. By applying an assimilation

window of 6 hours from the simulation start, the artificial volcanic eruption has not terminated, thus, the latest emissions from the nature run volcanic eruption are not considered in the analysis. This leads to a reduced pcc for the 6 hour assimilation window test case. The assimilation window of 12 hours ends four hours after the termination of the artificial volcanic eruption. The reduced pcc exhibits that this short assimilation window is not sufficient in order to analyze the correct emission profile. Thus, with increasing time after the end of the assimilation window, the pcc decreases to approx. 0.88 on 15 April 2010 and

approx. 0.92 on 29 April 2010. The high pcc values indicate that the assimilation of column mass loading to estimate volcanic ash emissions succeeds to retrieve the horizontal extent of the volcanic ash cloud. However, the pattern correlation coefficient is a measure for volcanic ash column mass loading above and below the chosen threshold. It does not measure differences in the strength of volcanic ash column mass loading above the threshold.

Increasing the assimilation window length (i.e. taking later observations into account) increases the pattern correlation coeffi-

cient on both days. The analysis suggests that for the respective test cases an assimilation window of 18 hours, that is 10 hours after the artificial eruption terminated, is sufficient for ESIAS-chem to analyze the exact location of the volcanic ash cloud as

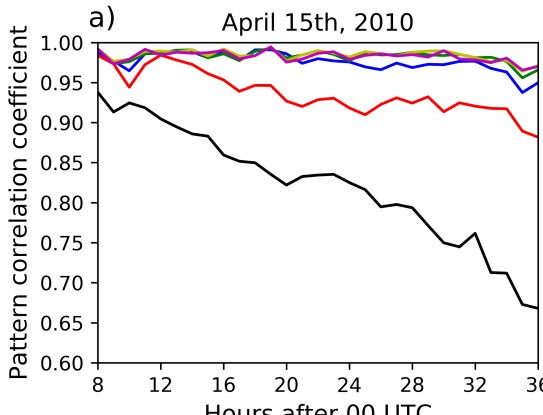
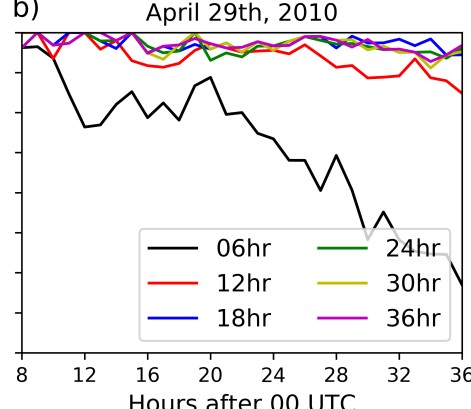

**Figure 6.** Pattern correlation coefficient defined by (7) for the eruption on (a) 15 April 2010 and on (b) 29 April 2010. The different lines indicate different assimilation window lengths from 6 hours (gray) to 36 hours (magenta) as defined by Fig. 3.

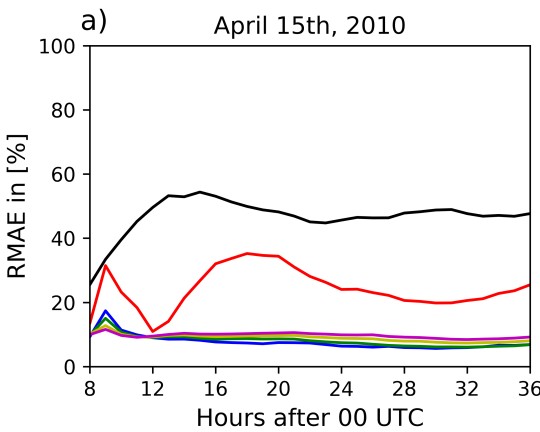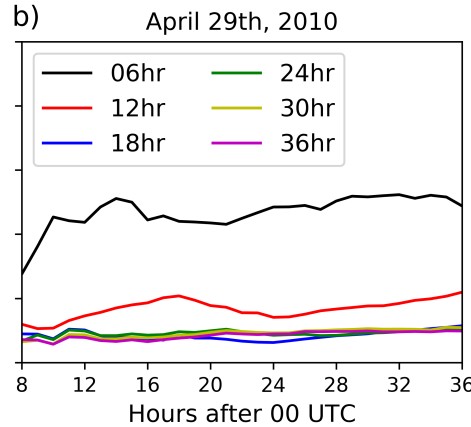

**Figure 7.** Relative mean absolute error of column mass loading defined by (9) for the eruption on (a) 15 April 2010 and on (b) 29 April 2010. The different lines indicate different assimilation window lengths from 6 hours (gray) to 36 hours (magenta) as defined by Fig. 3.

observed from space leading to a pcc value that remains high (> 0.95) throughout the full analysis time period of 36 hours. Fig. 6 demonstrates that the inversion system is able to accurately analyze the horizontal extent of the volcanic ash cloud.

Fig. 7 shows the RMAE for volcanic ash column mass loading on both days. The RMAE is relatively constant for the duration

of the simulations, except for the 12 hours assimilation window case in Fig. 7a. At the end of the simulation time at 36 hours, the test cases with longer assimilation windows ($\geq$ 18 hours) show a RMAE of the order of 10 % for both days. These low values show the good performance of the analysis for these assimilation window lengths with respect to the nature run. In principle, Fig. 7 corroborates the same findings that are analyzed for the pattern correlation coefficient, i. e. increasing the assimilation window length decreases the error of the analysis ensemble mean. For both days and meteorological circulation patterns, an

assimilation window of 18 hours is sufficient to reduce the RMAE to a value of approx. 10 % for column mass loading values above $0.2\,\mathrm{g\,m^{-2}}$. On 15 April 2010, assimilation windows larger than 24 hours result in a slightly higher RMAE than the analysis using an assimilation window of 18 hours. This is a result of the convergence of volcanic ash in the upper troposphere south of Norway around 24 hours after the simulation has started (not shown). Thus, additional observations at later times do not contribute significant information to the inversion system. In summary, Fig. 7 proves that the inversion system is able to

analyze the distribution of volcanic ash column mass loading properly for weak and strong wind conditions.

The above analysis focuses on the comparison of the nature run and the ensemble mean with respect to column mass loading of volcanic ash. Thus, it does not provide any information about the vertical distribution of volcanic ash. The ability of ESIAS-chem to infer vertical profiles of volcanic ash is given in Fig. 8, which displays the relative mean absolute error of the volcanic ash concentrations above $10\,\mathrm{\mu g\,m^{-3}}$. The RMAE of the volcanic ash concentrations decreases by increasing the assimilation

window length, which is especially visible for 29 April 2010. On both days, an assimilation window of only 6 hours results in a RMAE larger than 100 %. Therefore, this test case is not shown in Fig. 8. The RMAE for the 12 hour assimilation window

test case show a spike at 12 UTC. This results from the insufficient estimation of the upper part of the eruption column in the second explosive phase of the eruption (cf. Fig. A2 in the appendix). This error smoothed out in the subsequent hours of the simulation. On average, the RMAE reduces to about 20 % on both days for assimilation windows larger than 18 hours, which shows the good performance of the ESIAS-chem analysis not only in terms of column mass loading but also in terms of the vertical distribution of the volcanic ash in the atmosphere.

### 3.2.2 Emission profile

As an example, the analysis results using an assimilation window of 24 hours are investigated in more detail. This test case is chosen as the previous analysis showed the good performance of the 24 hour assimilation window experiments. Further, an assimilation window of 24 hours is a reasonable choice for either analysis of longer lasting volcanic eruptions or an operational use. The analyzed ensemble mean emission profiles for other assimilation window lengths are shown in the Appendix A along with the relative error. Fig. 9 and Fig. 10 display the profile of (a) the nature run emissions, (b) the ensemble mean emissions, (c) the relative error of the ensemble mean

$$RE = \frac{\overline{x} - y}{max(y)}, \tag{12}$$

and (d) the relative ensemble standard deviation

$$\sigma_{rel} = \frac{\sigma_x}{max(y)}, \tag{13}$$

for the 24 hour assimilation window experiments on 15 April and 29 April 2010. Herein, $\overline{x}$ and $y$ are the ensemble mean and nature run emissions, respectively, and $\sigma_x$ is the ensemble standard deviation.

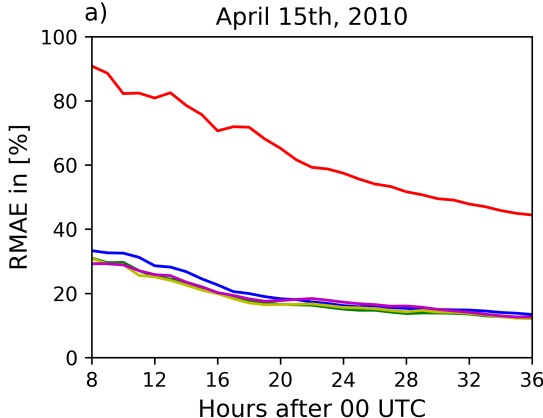
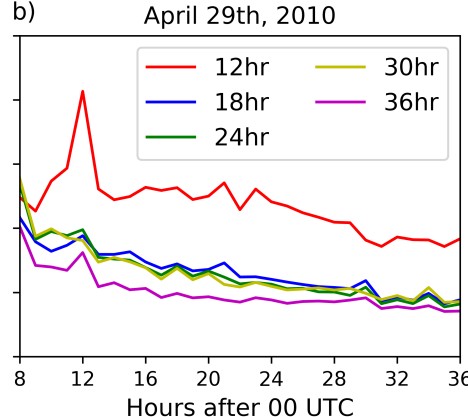

**Figure 8.** Relative mean absolute error of the volcanic ash concentrations defined by (9) for the eruption on (a) 15 April 2010 and on (b) 29 April 2010. The different lines indicate different assimilation window lengths from 12 hours (red) to 36 hours (magenta) as defined by Fig. 3.

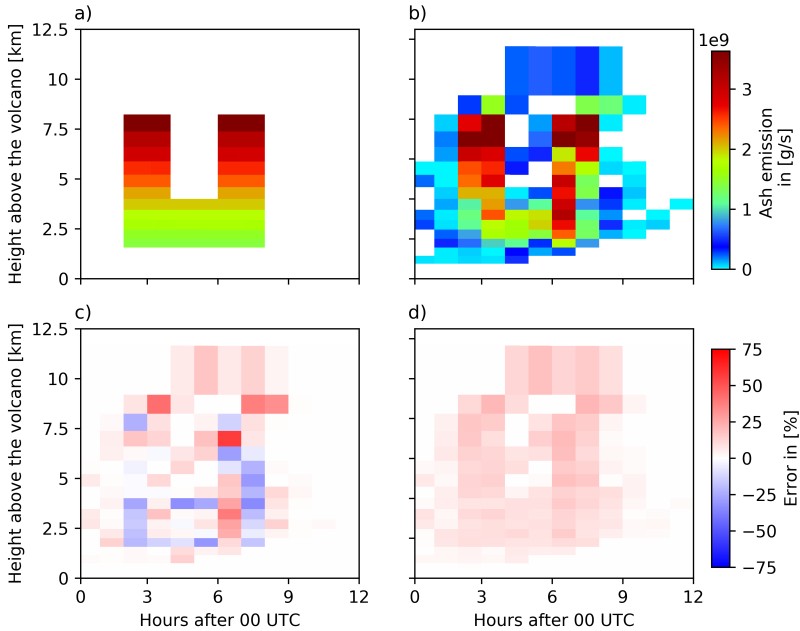

**Figure 9.** Comparison of the emission profiles of the nature run and ensemble mean for 15 April 2010. The figure shows (a) the nature run emission profile and (b) the ensemble mean emission profile. For comparison, (c) the relative error of the ensemble mean and (d) the relative ensemble standard deviation are illustrated.

The total nature run emissions on both days ($4.25 \cdot 10^8$ tons and $4.30 \cdot 10^8$ tons on 15 April and 29 April, respectively) are
well captured by the analyzed total emissions with mean emissions of $4.60 \cdot 10^8$ tons and $4.10 \cdot 10^8$ tons, respectively, and standard deviations of $3.67 \cdot 10^7$ tons and $3.47 \cdot 10^7$ tons. The relative error of the total emitted volcanic ash is 7.7 % and 4.7 %, respectively. On 15 April 2010, the analyzed emission profile of the ensemble mean shows the two explosive eruptions of the nature run emission profile with the correct height of the maximum emissions at the right time (Fig. 9b). Even though the ensemble mean shows a vertically and temporally smoothed emission profile, the false emissions are low with respect to the
maximum emissions. The relative error of the ensemble mean emissions is of the order of 10 %–20 % for most emission times and heights (cf. Fig. 9c) and therefore, the results are similar to the analysis presented before. The relative ensemble standard deviation is of the same order as the relative error of the ensemble mean emissions, indicating a reasonable ensemble spread. The analyzed emission profile of the ensemble mean on 29 April 2010 (Fig. 10b) however shows strong deviations from the nature run emission profile (Fig. 10a). Although the highest level emissions of the nature run emission profile at 8 km height
are well captured by the ensemble mean, at lower levels no distinction between the two explosive eruptions is obtained. In comparison to the analyzed emissions on 15 April 2010, on 29 April 2010 the analyzed emissions of the ensemble mean are more uniform in time and height. Thus, large errors in both directions can be identified: negative errors during the explosive

eruptions at around 03 UTC and 07 UTC indicate an underestimation of the emissions, while positive errors outside the two explosive eruptions indicate an overestimation of the emissions. This diffusion effect reflects the typical challenge of solving

ill–posed problems in reconstructing sharp spatial and temporal gradients. Nonetheless, the height and final time of the eruption are well analyzed by the ensemble mean on both days, which is basically a result of including no-volcanic ash observations. The analyzed emission profile on 29 April 2010 shows the limits of the ESIAS-chem approach. While the volcanic ash column mass loading have only low errors, the emission profile shows large deviations up to 60 % and more (Fig. 10c). The ensemble standard deviation of the emission profile (Fig. 10d) is lower than the relative error of the ensemble mean and ranges around

20 %. The results indicate that on 29 April 2010 the mixing of volcanic ash in the atmosphere is too effective, which prohibits a proper estimate of volcanic ash emission profiles. However, the previous results show that even though the volcanic ash emission profile could not be properly estimated by the system on 29 April 2010, the vertical and horizontal distribution of volcanic ash in the atmosphere is fairly represented by the ensemble mean.

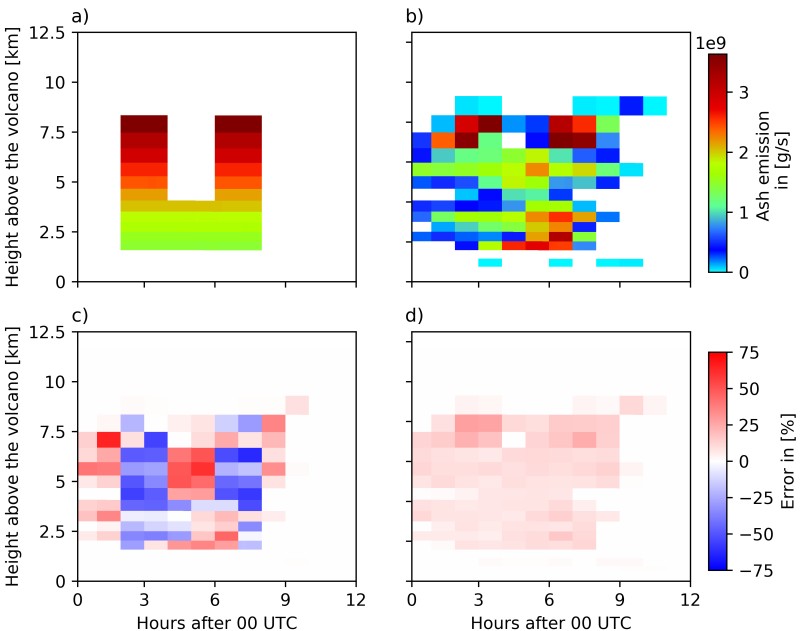

**Figure 10.** Comparison of the emission profiles of the nature run and ensemble mean for 29 April 2010. The figure shows (a) the nature run emission profile and (b) the ensemble mean emission profile. For comparison, (c) the relative error of the ensemble mean and (d) the relative ensemble standard deviation are illustrated.

### 3.2.3 Probability analysis

The proper analysis of high volcanic ash concentrations in the atmosphere as well as their forecast accuracy are of great importance for air safety advisory services. Yet, only the ability of ESIAS-chem to provide reasonable estimates of vertically resolved volcanic ash forecasts and analysis is shown. Thus, in this section the probability estimate of the analysis ensemble for the volcanic ash emissions and the resulting concentrations remains to be discussed. Fig. 11 shows the histogram of the relative emission factor for different assimilation window lengths for the test case on 15 April 2010 as given by the analysis

ensemble. The relative emission factor is calculated for each time-height combination $(t, k)$ of the emission profile by dividing the emission rate of each member of the analysis ensemble $ER_{t,k}^{(i)}$ by the respective nature run emission rate $ER_{t,k}^{NR}$

$$relEF_{t,k}^{(i)} = \frac{ER_{t,k}^{(i)}}{ER_{t,k}^{NR}}. \tag{14}$$

Thus, emissions in the analysis ensemble that are temporally or vertically outside the nature run emission profile are not considered. The calculation of the histogram in Fig. 11 includes all emissions and four different subset of emission strengths

(the strongest 50 %, 25 %, and 10 % emissions). The relative emission factors for the 12 hour assimilation window test case tend to underestimate the emissions of the nature run (Fig. 11a and Fig. 11d). By increasing the assimilation window length, the histograms peak around factor 1, while the occurrences of underpredicting the nature run emission rates diminish. A relative emission factor of 1 indicates a good match of the analyzed and nature run emission rates. This improvement by increasing the assimilation window length is especially true for the top 10 % emission rates in Fig. 11d.

Fig. 12 shows the histograms of the relative emission factors for the analysis on 29 April 2010. In general, the analysis tends to underestimate the emission rates as was previously discussed in Sect. 3.2.2. This results in a bias toward too small relative emission factors in the histograms. However, by increasing the assimilation window length, the underestimation of the emission rates by the analysis ensemble reduces. For the strongest 25 % of the emission rates, assimilation windows longer than 18 hours show a second maximum at a relative emission factor of 1 (Fig. 12c). These test cases also show a lower rate of underprediction

for the top 10 % emission rates (Fig. 12d). Thus, the results suggest that the reliability of the ensemble to analyze the strong emission rates in the upper emission plumes increases with increasing assimilation window length for both meteorological conditions, yet with different significance.

The accuracy of the probabilistic prediction of volcanic ash concentrations by the ensemble is measured by the Brier score (cf. Sect. 2.2). The Brier score is shown in Fig. 13 for each hour and for all assimilation window lengths. The Brier score

for assimilation windows greater equal 18 hours shows a low value around 0.15 on both analysis days, which is constant over time. Shorter assimilation windows have larger Brier score values that increase with simulation lead time. This increase of the Brier score for short assimilation windows is caused by insufficient estimates of the volcanic ash emissions, which lead to errors in the resulting volcanic ash concentrations as compared to the nature run. Thus, with increasing forecast time, the volcanic ash concentrations are attributed more and more to different classes used for the calculation of the Brier score.

This reduces the underlying probability and increases the Brier score. With increasing time after the volcanic eruption, the volcanic ash concentrations reduces due to dispersion and deposition. Lower volcanic ash concentrations have larger errors

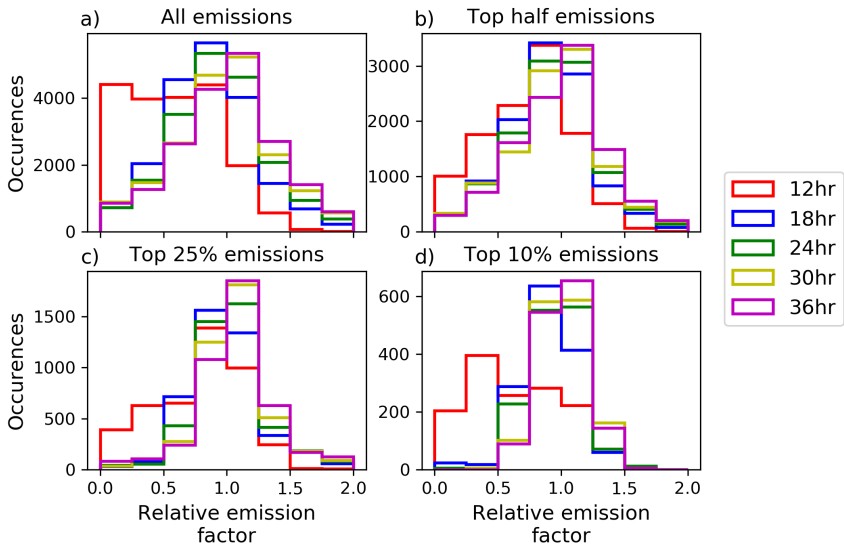

**Figure 11.** Histogram of the relative emission factors for different assimilation window lengths for the test case on 15 April, 2010 with colour codes as in Fig.3. The relative emission factor is calculated according to (14). The histograms are shown for (a) all emission rates, (b) the top half, (c) the top 25 %, and (d) the top 10 % emission rate.

(not shown) meaning that ESIAS-chem is less able to predict these low concentrations with high confidence. Especially for shorter assimilation window lengths, ESIAS-chem is not able to estimate the emission profile properly. Thus, the corresponding volcanic ash is emitted into false layers or at false times leading to larger errors in the probabilistic forecast.

As an example, the following analysis is aiming to assess the confidence of the ensemble prediction of volcanic ash using the 24 hour assimilation window experiment. Fig. 14a compares the probability of volcanic ash column mass loading exceeding $2 \, \mathrm{g \, m^{-2}}$ on 16 April 2010, 00 UTC. Additionally, the nature run's volcanic ash column mass loading contours for 0.5, 1, and $2 \, \mathrm{g \, m^{-2}}$ are overlaid by blue lines. On 15 April 2010, wind conditions are favorable for volcanic ash to disperse rapidly. Thus, the area containing high volcanic ash column mass loading covers only a small region above South-Sweden. The ensemble

predicts a probability of more than 90 % for high volcanic ash column mass loading in this area. A small probability of about 20-30 % of volcanic ash column mass loading exceeding the threshold of $2 \, \mathrm{g \, m^{-2}}$ is also predicted above the North Sea, where nature run's volcanic ash column mass loading exceeds $1 \, \mathrm{g \, m^{-2}}$. Fig. 14b shows the vertical cross-section along the red line in Fig. 14a, where the shading shows the probability of volcanic ash exceeding $500 \, \mathrm{\mu g \, m^{-3}}$. Nature run's volcanic ash concentrations are displayed by isolines of 250, 350, and $500 \, \mathrm{\mu g \, m^{-3}}$. As the dispersion of volcanic ash leads to low volcanic

ash concentrations on 16 April 2010, 00 UTC, the threshold of $500 \, \mathrm{\mu g \, m^{-3}}$ for calculating the exceedance probability was

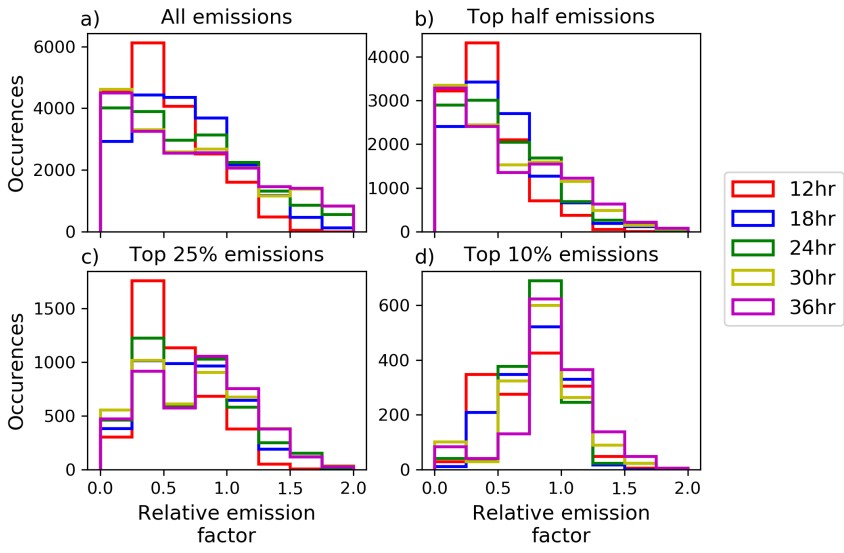

**Figure 12.** Histogram of the relative emission factors for different assimilation window lengths for the test case on 29 April 2010, with colour codes as in Fig.3. The relative emission factor is calculated according to (14). The histograms are shown for (a) all emission rates, (b) the top half, (c) the top 25 %, and (d) the top 10 % emission rate.

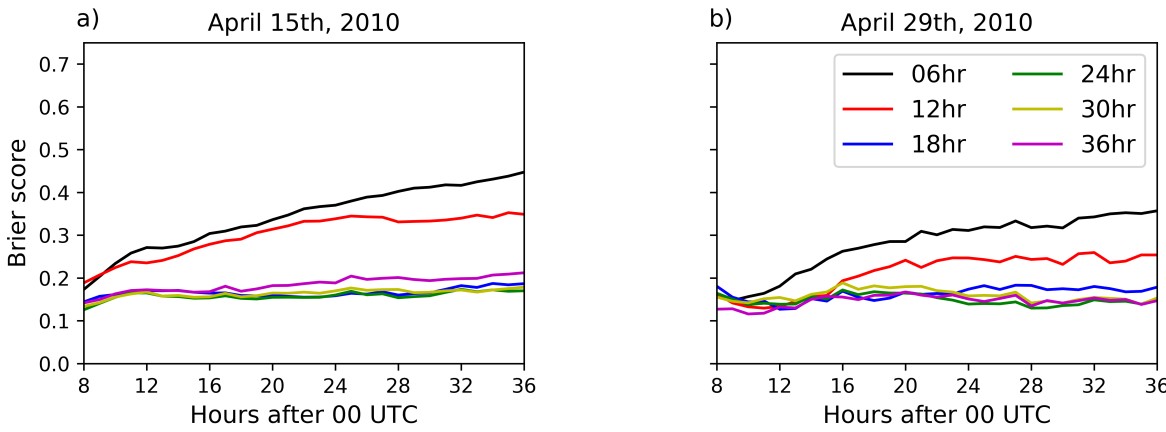

**Figure 13.** Brier score as calculated by (10) for each hour and all assimilation window lengths for (a) 15 April and (b) 29 April 2010.

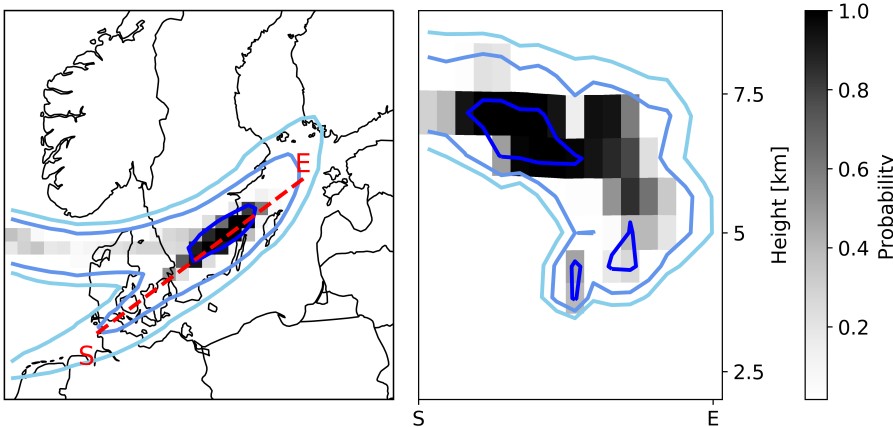

**Figure 14.** Probability maps on 16 April 2010, 00 UTC, from the 24 h assimilation window test case. a) Probability of the analysis ensemble for volcanic ash column mass loading exceeding $2 \, \mathrm{g \, m^{-2}}$ (shaded areas). Contour lines of nature run volcanic ash column mass loading for 0.5, 1, and $2 \, \mathrm{mg \, m^{-2}}$ are given by blue lines. b) Vertical contours of the probability of the analysis ensemble of volcanic ash concentration exceeding $500 \, \mathrm{\mu g \, m^{-3}}$ along the red dashed line in a). Contour lines of nature run volcanic ash concentrations for 250, 350, and $500 \, \mathrm{\mu g \, m^{-3}}$ are shown by blue lines.

chosen instead of using the official threshold of $2 \, \mathrm{mg \, m^{-3}}$ (Prata and Prata, 2012, and references therein). The nature run's volcanic ash concentrations of more than $500 \, \mathrm{\mu g \, m^{-3}}$ at about $7 \, \mathrm{km}$ height are well represented by the ensemble with high probability ($> 90 \, \%$). Two other locations in this vertical cross-section show nature run's volcanic ash concentrations above $500 \, \mathrm{\mu g \, m^{-3}}$ at approx. $4 \, \mathrm{km}$ height in the center of the vertical cross-section and at approx. $5 \, \mathrm{km}$ height northeast of the
525 center (i. e. to the right in Fig. 14b). Even though the volcanic ash at $4 \, \mathrm{km}$ height in the center of the cross-section is covered from above by the elevated volcanic ash at $7 \, \mathrm{km}$ height, the ensemble predicts a $50 \, \%$ chance of volcanic ash exceeding the threshold at $4 \, \mathrm{km}$ height. This is remarkable, since only vertically integrated observations of volcanic ash are assimilated. The volcanic ash northeast of the center of the vertical cross-section (i. e. to the right in Fig. 14b) is predicted by only 20-30 % of the ensemble. The ensemble predicts this volcanic ash in this vertical column to be at a height of 6-7 $\mathrm{km}$ by a chance of more
than $70 \, \%$. This may be due to the lack of vertical wind shear that prevents the distinction of volcanic ash emitted at different height levels.

Fig. 15a shows the probability of volcanic ash column mass loading exceeding $2 \, \mathrm{g \, m^{-2}}$ as predicted by the ensemble on 30 April 2010, 12 UTC, i. e. 36 hours after the simulation start and 12 hours after the end of the assimilation window. Isolines of nature run's volcanic ash column mass loading for $0.5 \, \mathrm{g \, m^{-2}}$, $1 \, \mathrm{g \, m^{-2}}$, and $2 \, \mathrm{g \, m^{-2}}$ are also given by blue lines. A vertical
cross-section of the probability of volcanic ash concentration exceeding $2 \, \mathrm{mg \, m^{-3}}$ along the red line in Fig. 15a is shown in Fig. 15b. Even though the emission profile on 29 April 2010 was not well analyzed, the ensemble predicts the high volcanic ash concentration with a probability of more than $90 \, \%$. Fig. 15b shows a vertically tilted volcanic ash cloud. This suggests that only little vertical mixing occurred on 29. April 2010 in the displayed vertical cross-section. Thus, the falsely emitted volcanic

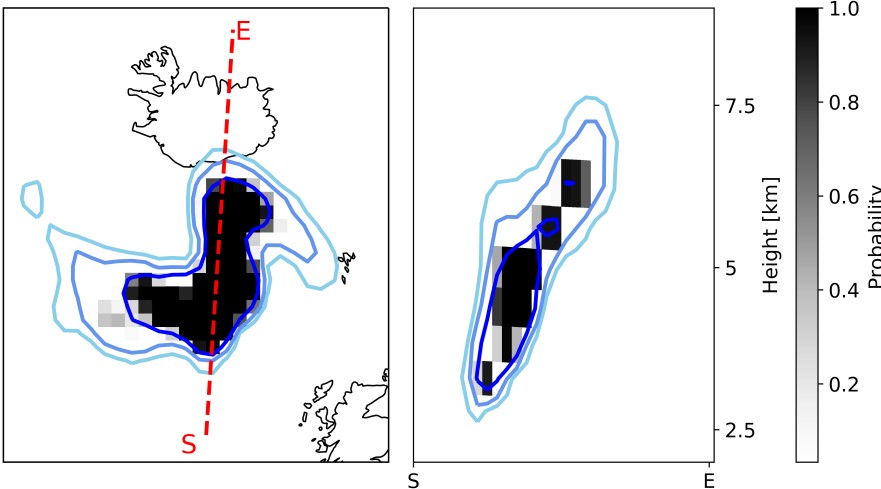

**Figure 15.** Probability map on 30 April 2010, 12 UTC, from the 24 h assimilation window test case. a) Probability of the analysis ensemble for volcanic ash column mass loading exceeding $2\,\mathrm{g\,m^{-2}}$ (shaded areas). Contour lines of nature run volcanic ash column mass loading for 0.5, 1, and $2\,\mathrm{g\,m^{-2}}$ are given by blue lines. b) Vertical contours of the probability of the analysis ensemble of volcanic ash concentrations exceeding $2\,\mathrm{mg\,m^{-3}}$ along the red dashed line in a). Contour lines of nature run volcanic ash concentrations for 0.5, 1, and $2\,\mathrm{mg\,m^{-3}}$ are shown by blue lines.

ash in the horizontally smoothed analysis emission profile leads to similar volcanic ash concentrations, which suggests that

horizontal mixing of volcanic ash happened. Hence, an exact estimation of the emission profile is generally impossible from column mass loading observations as different emission packages lead to similar volcanic ash concentrations and/or column mass loadings. However, the good performance in analyzing the vertical structure of the volcanic ash cloud is partly due to the perfect model / perfect meteorology assumption made in this study. The reliable estimate of the emission profile for the test case with strong wind shear suggest that the vertical structure of the volcanic ash is also sufficiently estimated under real

conditions, where meteorological forecast uncertainties impose a limiting factor to further improvements. This needs to be proved in the application to real volcanic eruptions.

## 4   Discussion and conclusions

In this study, a new method for estimating volcanic ash emissions and its uncertainty from column mass loading observations is developed. This new method is realized by the atmospheric chemical part of the Ensemble for Stochastic Integration of

Atmospheric Simulations (ESIAS-chem). The method comprises an ensemble-based particle smoother, which extends the assimilation window to include the latest observations available. The discrete-grid ensemble Nelder-Mead method (DENM) is developed in order to efficiently achieve a posterior ensemble representation of the time-dependent emission profile. The particle smoother approach enables to use the latest observations for the estimation of the emission profile within the whole

assimilation window, while consistancy with all observations within the time interval is enforced.

The system was applied in an idealized setup to a notional eruption of the Eyjafjalljökull volcano, Iceland, on 15 April and 29 April 2010 using a sub-Plinian type eruption with two short eruption plumes. Both days were characterized by different meteorological conditions. On 15 April 2010, strong winds were present at the volcano, while on 29 April 2010, winds were weak. In the identical twin experiments, different assimilation window lengths have been tested to investigate the influence of increasing observation sequences on the analysis quality. The main findings in this study are that

– the error of the analyzed column mass loading and volcanic ash concentrations by the ensemble mean is only 10 % on 15 April and 20 % on 29 April 2010,

– the total emitted mass of volcanic ash is reasonably well estimated by the analysis ensemble on both days,

– by increasing the assimilation window length, the ensemble performs increasingly better in analyzing the emission rates, especially for high emission rates in the upper part of the eruption column,

– on 15 April 2010, a second lower volcanic ash layer covered from above by the main volcanic ash cloud was predicted by about 50 % of the ensemble members.

Due to the identical twin approach, the presented investigation acts as a best case scenario for probabilistic volcanic ash assessments. The analysis is idealized in different ways: The uncertainties in meteorological fields (especially in winds) in model parameters (e. g. deposition velocity), and parametrizations (e. g. clouds) have been neglected. Further, the amount of

570 observational data is exceptionally large, with observations of the full domain every 6 hours. Thus, observations of ash-free areas allow for removing volcanic ash emissions from the analysis. The ability of ESIAS-chem to give reliable results for real volcanic eruptions using non-idealized meteorology and incomplete observations will be addressed in a follow-up study. Even though direct observations of volcanic ash columns were used in this study, ESIAS-chem is extremely flexible in terms of observational data. All kinds of data can be used the constrain the inversion method, such as samples of tephra fall out, if

available.

ESIAS-chem is designed to account for additional information on the emission profile, which may, for example, be obtained from radar or web cam observations (e. g., Arason et al., 2011). Thus, changes in the vertical or temporal resolution of the emission profile are applicable if suggested by observations without noteworthy modifications.

In this study, ESIAS-chem was challenged with highly variable volcanic ash emissions. The analysis has shown that ESIAS-

580 chem is able to provide good estimates of the volcanic ash concentration in the atmosphere as well as its forecast probability. Further, the emission profile was estimated reasonably well at least for the strong wind test case for assimilation window lengths greater than 18 hours. However, the ideal length of the assimilation window may depend on the current meteorological situation, most notable the vertical wind shear, and the availability of observational data. Thus, in applications to real volcanic eruptions the assimilation window should be as large as practicably possible to include a large number of observations linking

eruption time of particles with observation time.

The system shows high probability in estimating the vertical distribution of high volcanic ash concentration for both test dates.

Although the system lacks to estimate the true emission profile sufficiently well for weak wind conditions, the analysis of the probability of volcanic ash showed that its vertical distribution in the atmosphere is reliably predicted.

Besides volcanic ash eruptions, ESIAS-chem is applicable to a variety of emission scenarios, especially unexpected emission events like forest fires and mineral dust events. Therefore, it provides a fast and efficient model for source term estimation including uncertainty representation. In principle, the method can be adapted to multi-source emission scenarios. The enhanced need for compute resources of ESIAS-chem can partly be compensated by reducing the resolution of the emission profile. For the analysis of real volcanic ash emissions, it is intended to use a meteorological ensemble to account for additional uncertainties in wind fields, which is well applicable within the concept of ESIAS-chem. It is noted that ESIAS-chem is flexible in integrating other modules and is applicable to other atmospheric models as well.

*Code availability.* The analysis module used to calculate the volcanic ash emission estimate can be downloaded at http://doi.org/10.5281/zenodo.4736071 under a Creative Commons Attribution 4.0 International License.

# Appendix A: Comparison of emission profiles for all assimilation windows

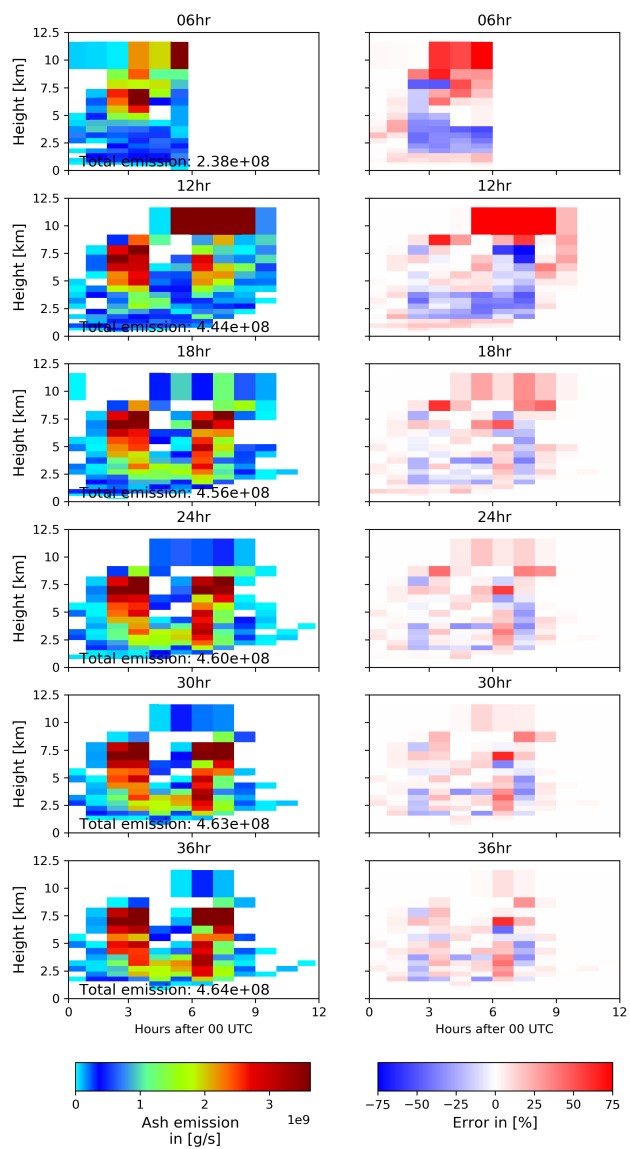

**Figure A1.** Emission profile of the ensemble mean (left panels) and its relative error to the nature run emission profile (right panels) for different assimilation window lengths on 15 April 2010.

*Author contributions.* PF designed the code extension necessary for this analysis. HE contributed mainly to the idea of ensemble estimation

of emission uncertainties. Main input regarding satellite observations and its uncertainty was given by ACL. All authors contributed equally

to the manuscript.

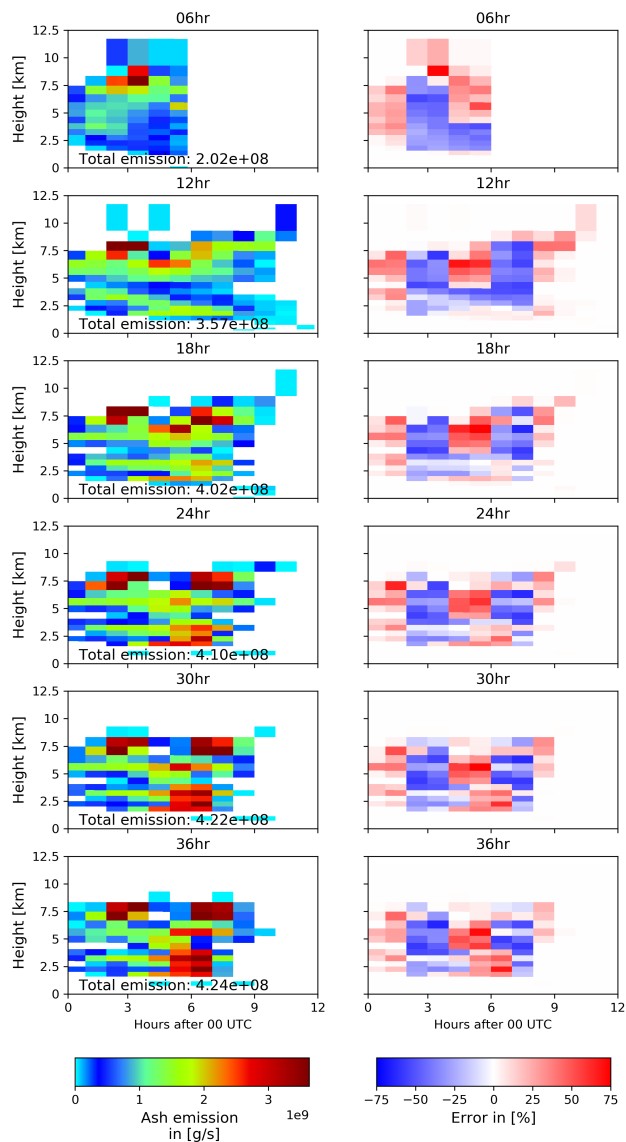

**Figure A2.** Emission profile of the ensemble mean (left panels) and its relative error to the nature run emission profile (right panels) for different assimilation window lengths on 29 April 2010.

*Competing interests.* The authors declare that they have no competing interests.

*Acknowledgements.* The authors gratefully acknowledge the Gauss Centre for Supercomputing e.V. (www.gauss-centre.eu) for funding this project by providing computing time through the John von Neumann Institute for Computing (NIC) on the GCS Supercomputer JUQUEEN
at Jülich Supercomputing Centre (JSC). The authors specially acknowledge Olaf Stein and Sebastian Lürs from JSC for their support of implementing the ensemble version of the EURAD-IM and improving its performance. This work is part of the Helmholtz program Earth System Knowledge Platform (ESKP) and was established in the Helmholtz Interdisciplinary Doctoral Training in Energy and Climate Research (HITEC) at Forschungszentrum Jülich and in the Graduate School of Geosciences (GSGS) at the University of Cologne.

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
