# Peer review of "Particle filter based volcanic ash emission inversion applied to a hypothetical sub-Plinian Eyjafjallajökull eruption using the Ensemble for Stochastic Integration of Atmospheric Simulations (ESIAS-chem) version 1.0"

_Geoscientific Model Development, 2021_

## Referee Comment (RC1)

**Review of**
*"Particle filter based volcanic ash emission inversion applied to a hypothetical sub-Plinian Eyjafjallajökull eruption using the chemical component of the Ensemble for Stochastic Integration of Atmospheric Simulations (ESIAS-chem) version 1.0"*
**by Franke et al.**

Review by Nina Kristiansen

**General comments**

The paper presents a methodology to estimate volcanic ash emissions as a function of time and altitude based on observations and modelled ensemble simulations. The main strength of the paper is that the methodology also gives an estimate of the uncertainty/errors in the estimated emissions, however this aspect could be more clearly demonstrated throughout the paper with a clearer description of how this uncertainty estimate is obtained. There is potential to make a few sections clearer and easier for the reader to follow, in particular the methodology section. The results shown are interesting and the figures clear. The paper is suitable for publication once the below comments have been addressed.

**Specific comments**

**Abstract**:

- "*The system validation addresses the special challenge of ash cloud height analyses in case of observations restricted to bulk column mass loading information, mimicking the typical case of geostationary satellite data.*" – unclear what is meant here, please rewrite.
- The abstract should more clearly say that you are using an "idealized situation with artificial observations from a model run" and that you use many observations of both ash and ash-free areas. It should also say that meteorological uncertainty is not included.
- L20: "*This situation, however, can be remedied by extending the assimilation window*". I am not sure this is true, as in the results section you show Figure 10 for the weak wind shear situation that the emission profile is not well estimated and you don't show a better estimation when including more observations (i.e. increasing the assimilation window >24 hrs) for this case.

**Section 1 Introduction:**

- Line 32: "*Typically, volcanic eruptions occur as sequences of emissions with highly varying ejection mass and height*". This might be correct for explosive eruptions but not necessarily for effusive or passive degassing.
- Line 33: You might want to include that radar observations also are uncertain and have limitations.
- Line 36: "*Statistical models are based on observational data from only a few, highly heterogeneous volcanic eruptions*". I don't think the word heterogenous is right here. The issue with the statistical methods by Mastin/Sparks is mainly that it is biased to larger eruptions (very small number of smaller eruptions were included in the empirical estimates), but I wouldn't say that the eruptions considered were 'heterogeneous'.

- Line 38: "*physical plume-scale models require vent and magma details, which are poorly known, and thus making these models highly uncertain.*" It might be more informative to include something on how sensitive the plume models are to the vent/magma details and the expected magnitude of errors associated with this. See the plume model intercomparison study by Costa et al 2016 (https://www.sciencedirect.com/science/article/abs/pii/S0377027316000366)
- Line 50: "*is the horizontally more complete picture of the volcanic ash extent*" unclear what is meant by more complete picture here – and more complete compared to what?
- Line 73: Would be good to include some further details of the advantages and limitations of each method which you mention. Also include more recent papers on data assimilation/insertion:

  Prata, A. T., Mingari, L., Folch, A., Macedonio, G., and Costa, A.: FALL3D-8.0: a computational model for atmospheric transport and deposition of particles, aerosols and radionuclides – Part 2: Model validation, Geosci. Model Dev., 14, 409–436, https://doi.org/10.5194/gmd-14-409-2021, 2021.

  Pardini, F., Corradini, S., Costa, A., Esposti Ongaro, T., Merucci, L., Neri, A., Stelitano, D., and de' Michieli Vitturi, M.: Ensemble-Based Data Assimilation of Volcanic Ash Clouds from Satellite Observations: Application to the 24 December 2018 Mt. Etna Explosive Eruption, Atmosphere, 11, 359, https://doi.org/10.3390/atmos11040359, 2020.

  Osores, S., Ruiz, J., Folch, A., and Collini, E.: Volcanic ash forecast using ensemble-based data assimilation: an ensemble transform Kalman filter coupled with the FALL3D-7.2 model (ETKF–FALL3D version 1.0), Geosci. Model Dev., 13, 1–22, https://doi.org/10.5194/gmd-13-1-2020, 2020.

  Fu, G.; Lin, H.X.; Heemink, A.; Lu, S.; Segers, A.; Velzen, N.V.; Lu, T.; Xu, S. Accelerating volcanic ash data assimilation using a mask-state algorithm based on an ensemble Kalman filter: A case study with the LOTOS-EUROS model (version 1.10), Geosci. Model Dev., 10, 1751–1766, 2017.

  Fu, G., Prata, F., Lin, H. X., Heemink, A., Segers, A., and Lu, S.: Data assimilation for volcanic ash plumes using a satellite observational operator: a case study on the 2010 Eyjafjallajökull volcanic eruption, Atmos. Chem. Phys., 17, 1187–1205, https://doi.org/10.5194/acp-17-1187-2017, 2017.

**Section 2:**

- I generally find the methodology section difficult to follow. There are many technical terms and abbreviations to keep track of, and the descriptions are sometimes not clear. Perhaps an extension of Figure 1 (the flow diagram) to include further steps and references to methods/terminology/naming conventions would help. I also suggest expanding the figure caption of Figure 1 to explain what the figure shows which makes it easier for the reader to return to this figure later while reading subsequent sections.
- L 98: "*Stohl et al. (2011) and Kristiansen et al. (2015) aiming to estimate the optimal emission profile but not its uncertainty*". I think it is fair to say that this work did provide some uncertainty estimates (and included uncertainty in both the a priori, the observations and the model input – though this could of course be improved). In figure 3 of Stohl et al the uncertainty reduction from the a priori via the inversion to the a posteriori is shown. This shows how much influence the observations had, and which parts of the emissions were well constrained by the

observations and which were less constrained, and therefore is a form of uncertainty estimate. I do appreciate that what you are aiming to provide is different (but I still don't quite understand how the uncertainty is estimated!) but it would be good to include some more details here how your uncertainty estimate differ from this to make it clear.

**Section 2.2:**

- Some further comments on the advantages and disadvantages of the Nelder-Mead method would be good to include. For example, mentioning that the reason the method is suitable for "discontinuous"/"spiky"/"noisy" problems is because it does not use derivatives, but also that it doesn't use a convergence theory and doesn't necessarily find the minimum function value (but rather an 'improvement') – that is the key difference to the method used by Stohl etc.
- "*The minimization was performed in NN0, which has been found to be more effective than the minimization in RN*" –to help the reader please directly spell out what this means. I think you mean in model space rather than in observation space. And if this is the case is this more efficient because the number of ensemble members is smaller than the number of observations?
- How is the initial simplex determined for each ensemble member? You say later it is arbitrary but some more details here would be useful.
- It would also be nice to see some comment on computational effort (i.e. run time) for this system.

**Section 2.3:**

- L 140: "*It is noted that in the particle filter method no assumptions of the error statistics of the model state and the observations were made.*" I don't understand how this relates to the uncertainty estimate you apply in the results section where you assume a 40% uncertainty on the observations….
- L 145 "*the ensemble members with high weights are duplicated and perturbed, replacing ensemble members with vanishing weights.*" I don't quite understand how this works in practice with the unit ensemble members... please expand on this part.

**Section 2.4**

- L 150: "*In order to account for meteorological uncertainties, ESIAS-chem is capable to be coupled with ensembles of meteorological field*". But you have not used met ensembles here? Please clarify in text if this is only a possible extension.
- L 179: "*Finally, a particle filter step is applied. The weights, which result from the filtering step, are applied to the optimized emission profiles.*" Please provide some more details of how the "ensemble mean" is constructed (a term which you use in Section 3) and then also the uncertainty estimates based on the ensembles. I understand that each ensemble member simulates the emission released by a single emission package for an individual time step and height interval and then weighted by the likelihood in the particle filter step. How do you construct the ensemble mean from this? Are the ensemble members still associated with emissions from a single emission package but after the filtering with an amount of ash (rather than a unit release)?

**Section 3:**

- L187: "*Given the identical twin assumption the experiment is then to be made realistic in all other respects, as the two different weather conditions on our case*". This sentence is not entirely clear. I think you refer to the "twins" as the two different weather conditions. It might be better for the reader if you say this in the first sentence of this paragraph.
- L190: Is EURAD-IM an online or offline model? Please clarify and which meteorological data are used as driver/lateral boundary conditions.
- L203: "*The uncertainty of volcanic ash column mass loading observations is about 40 % (Kristiansen et al., 2015, and references therein)*": Some better references here might be:
  - L. Clarisse, F. Prata: Chapter 11 - Infrared Sounding of Volcanic Ash Editor(s): Shona Mackie, Katharine Cashman, Hugo Ricketts, Alison Rust, Matt Watson, Volcanic Ash, Elsevier, 2016, Pages 189-215, ISBN 9780081004050, https://doi.org/10.1016/B978-0-08-100405-0.00017-3. (https://www.sciencedirect.com/science/article/pii/B9780081004050000173)
  - Kylling, A.; Kahnert, M.; Lindqvist, H.; Nousiainen, T. Volcanic ash infrared signature: porous non-spherical ash particle shapes compared to homogeneous spherical ash particles. Atmos. Meas. Tech. 2014, 7, 919–929. 144.
  - Western, L.; Watson, I.; Francis, P. Uncertainty in two-channel infrared remote sensing retrievals of a well-characterised volcanic ash cloud. Bull. Volc. 2015, 77, 67.

- Equation 7: Here you use an observation error 40% of the observation value. It might be worth commenting here that when using real observations (instead of synthetic as in your case) then a better approach would be to use the retrieval uncertainty estimate for each single satellite pixel, and not a fixed uncertainty value..
- L219+223: "*The length of the assimilation window influences the performance of the data assimilation algorithm due to the influence of vertical and horizontal mixing and vertical wind shea*r." And "*Certainly, by increasing the assimilation window length the observations include more information, as the residence time of volcanic ash in the atmosphere is increased*". The wording here is a bit strange. The residence time of volcanic ash in the atmosphere doesn't increase with increased assimilation window. As the assimilation window increases the number of observations which are assimilated increases… Please rewrite.
- Page 11-12: There is a lot of jumping between the pcc and RMAE. Might be worth explaining first one and the results then the other one and the results.
- L265: 10 μgm−3 seems like a very low concentration threshold (considering the aviation thresholds starting at 200 ug/m3). What was the reasoning behind this threshold, and do the results change if the threshold is higher?
- L282: "*The analysis suggests that for the respective test cases an assimilation window of 18 hours, that is 10 hours after the artificial eruption terminated, is sufficient for ESIAS-chem to analyze the exact location of the volcanic ash cloud…*" This is a nice result (and backed up by the RMAE later) which I think you should put in the abstract.
  You could also here refer to work by Fu et al who also analysed this "effective duration" (i.e. the required temporal cycle to obtain improved forecasts). Fu et al. 2015, 2016, 2017 report between 6 and 24 hours.
  - Fu, G.; Lin, H.; Heemink, A.; Segers, A.; Lu, S.; Palsson, T. Assimilating aircraft-based measurements to improve forecast accuracy of volcanic ash transport, Atmos. Environ., 115, 170–184, 2015.

- o   Fu, G.; Heemink, A.; Lu, S.; Segers, A.; Weber, K.; Lin, H.X. Model-based aviation advice on distal volcanic ash clouds by assimilating aircraft in situ measurements, Atmos. Chem. Phys., 16, 9189–9200, 2016.
  - o   Fu, G., Prata, F., Lin, H. X., Heemink, A., Segers, A., and Lu, S.: Data assimilation for volcanic ash plumes using a satellite observational operator: a case study on the 2010 Eyjafjallajökull volcanic eruption, Atmos. Chem. Phys., 17, 1187–1205, https://doi.org/10.5194/acp-17-1187-2017, 2017.
  - o   Fu, G.; Lin, H.X.; Heemink, A.; Lu, S.; Segers, A.; Velzen, N.V.; Lu, T.; Xu, S. Accelerating volcanic ash data assimilation using a mask-state algorithm based on an ensemble Kalman filter: A case study with the LOTOS-EUROS model (version 1.10), Geosci. Model Dev., 10, 1751–1766, 2017b.

- L305: "*good performance of the ESIAS-chem analysis not only in terms of column mass loading but also in terms of the vertical distribution of the volcanic ash in the atmosphere*". It might be worth re-iterating here that getting a good performance for the concentrations are possibly strongly affected by the use of "perfect meteorology" and that such good results are not expected using real observations.
- L306: I was surprised why the 18 hours assimilation windows wasn't chosen here over the 24 hours as the 18 h seems to show equally good results up until now.
- You first show the validation using pcc and RMAE (Figs 6-8) for all assimilation time windows, and then the results for one of the assimilation windows (figure 9-10). I would prefer it the other way around so that I can see what the estimated emission profile looks like (for one of the assimilation windows) before it is validated and tested against the other assimilation windows. Also, because the main strength of your method is giving an uncertainty estimate for the emission profile this should be the focus of the results.
- Figure 9 and 10: It would be interesting to see "b" figure for all assimilation windows, to see how the emission profile changes as you assimilate more and more observations, and how the estimated uncertainty (c, d figures) also changes when including more observations.
- L322: In the abstract you say that for the strong wind shear condition the estimated emissions have "up to an error of only 10 %" but here you say relative errors are around 10-20 %. Also, in the abstract you say that in a situation with little wind shear the errors are "higher", while here you say up to 60% and more. I would change the abstract to give the same numbers as here.
- L337: "*The results indicate that on 29 April 2010 the mixing of volcanic ash in the atmosphere is too effective*". With much less wind shear on 29 April it seems that the problem isn't too much mixing but that the emissions at different altitudes and times are transported in a similar way and thus cannot be easily separated by the assimilation.
- L340: "*However, the previous results show that even though the volcanic ash emission profile could not be properly estimated by the system on 29 April 2010, the vertical and horizontal 340 distribution of volcanic ash in the atmosphere is fairly represented by the ensemble mea*n.". This is a little worrying. The fact that the pcc and RMAE give such good scores even with such a "smooth" emission profile after the assimilation which deviates strongly from the nature run emissions ("truth")... it does make me question whether the pcc and RMAE are appropriate statistical measures to be used here… But it might be more to do with the fact that with little wind shear many different emission profiles can equally well give a best fit with the observations and the fact that the Nelder-Mead method doesn't necessarily find the minimum only an "improvement" as previously mentioned. I think the point that the emissions are not well estimated, but that the simulated concentrations and column loadings still fit well with the nature run would be a little bit more explored and discussed.

- L371: "*Thus, ESIAS-chem demonstrates to estimate the vertical distribution of volcanic ash in the atmosphere on both simulation days with a high accuracy*." A probability of 90% from the ensemble does not mean that the simulation is of high accuracy.

**Section 4 Conclusions**

- This reads a bit more like a discussion and future outlook. I would rename this to "discussion" and include another section for the Conclusions which summarizes the key results you have shown with a few bullet points.

**Technical corrections**

Line 33: "form radar" – change to "from radar"

Line 285: "extend" – change to "extent"

---

## Author Response (AR1)

Response to Nina Kristiansen on her review:

We colored our response to Nina Kristiansen in blue.

General comments

The paper presents a methodology to estimate volcanic ash emissions as a function of time and altitude based on observations and modelled ensemble simulations. The main strength of the paper is that the methodology also gives an estimate of the uncertainty/errors in the estimated emissions, however this aspect could be more clearly demonstrated throughout the paper with a clearer description of how this uncertainty estimate is obtained. There is potential to make a few sections clearer and easier for the reader to follow, in particular the methodology section. The results shown are interesting and the figures clear. The paper is suitable for publication once the below comments have been addressed.

We thank Dr. Kristiansen for her detailed and helpful review. We are confident that our modifications are in compliance with her expectations, which would certainly improve our manuscript.

Specific comments

Abstract:

- "The system validation addresses the special challenge of ash cloud height analyses in case of observations restricted to bulk column mass loading information, mimicking the typical case of geostationary satellite data." – unclear what is meant here, please rewrite.
  We have rewritten this phrase: "Thus, the proposed system addresses the special challenge of analyzing the vertical profile of volcanic ash clouds given only column mass loading data as retrieved by geostationary satellite imagery."
- The abstract should more clearly say that you are using an "idealized situation with artificial observations from a model run" and that you use many observations of both ash and ash-free areas. It should also say that meteorological uncertainty is not included.
  We added this information to the abstract. The phrase concerning identical twin experiments now reads: "As initial validation of ESIAS-chem, the system is applied to simulated artificial observations of both ash-contaminated and ash-free atmospheric columns using identical twin experiments. Thus, in this initial performance test the underlying meteorological uncertainty is neglected."

L20: "This situation, however, can be remedied by extending the assimilation window". I am not sure this is true, as in the results section you show Figure 10 for the weak wind shear situation that the emission profile is not well estimated and you don't show a better estimation when including more observations (i.e. increasing the assimilation window >24 hrs) for this case.

Thank you for this comment. Indeed, extending the assimilation window does not guarantee a better performance. Anyhow, even in the case of weak wind shear, the confidence of the analysis ensemble on the time and height of the volcanic ash emissions increases with increasing assimilation window length. This has been addressed in the new Fig. 11 where you see a histogram of the relative emission factor of the analysis for the weak wind shear test case. We have changed the sentence: „In case of increasing wind shear, the performance of the analysis may benefit from

extending the assimilation window, in which new observations potentially contribute valuable information to the analysis system. "

Section 1 Introduction:

- Line 32: "Typically, volcanic eruptions occur as sequences of emissions with highly varying ejection mass and height". This might be correct for explosive eruptions but not necessarily for effusive or passive degassing.
  We agree. We have added "explosive" before volcanic eruptions in this phrase.
- Line 33: You might want to include that radar observations also are uncertain and have limitations.
  Thank you for this comment. We added this information to the sentence. It now reads: "Only limited observations of volcanic ash emission parameters are available (e.g. eruption plume heights retrieved from radar measurements, Arason et al., 2011), which are affected by their specific uncertainties and limitations, e. g. by orographic shielding."
- Line 36: "Statistical models are based on observational data from only a few, highly heterogeneous volcanic eruptions". I don't think the word heterogenous is right here. The issue with the statistical methods by Mastin/Sparks is mainly that it is biased to larger eruptions (very small number of smaller eruptions were included in the empirical estimates), but I wouldn't say that the eruptions considered were 'heterogeneous'.
  Thank you for this helpful comment. We have added this information. The point we wanted to highlight is the large variance of the eruption rate given a specific plume height (Fig. 1 in Mastin et al. 2009). We agree, that "heterogeneous" is insufficient to explain this relation. Thus, we have changed the sentence to: "Statistical models base on observational data from historic volcanic eruptions, which are sparse and show a large variance in eruption rate for given plume heights. For example, Mastin et al. (2009) calculated an uncertainty by a factor of four in estimating the emission rate for a plume height of 25 km using their statistical model."
- Line 38: "physical plume-scale models require vent and magma details, which are poorly known, and thus making these models highly uncertain." It might be more informative to include something on how sensitive the plume models are to the vent/magma details and the expected magnitude of errors associated with this. See the plume model intercomparison study by Costa et al. 2016 (https://www.sciencedirect.com/science/article/abs/pii/S0377027316000366)
  As for the statistical models, we have added some information about the uncertainty of the plume-scale models. The sentence has been expanded: "Physical plume-scale models require orographic details of the volcano (e. g. crater size) but also meteorological fields and parameters (e. g. wind entrainment coefficients), which are often poorly known and render these models highly uncertain. Costa et al. (2016) identified the wind entrainment coefficient as main source of uncertainty leading to up to two orders of magnitude differences for the estimation of mass eruption rates for weak volcanic eruptions. In their analyses of the eruptions of the Eyjafjallajökull, Iceland, in 2010 and Grímsvötn, Iceland, in 2011, Woodhouse et al. (2015) found a comparable range of uncertainty depending on the choice of the wind entrainment coefficients."
- Line 50: "is the horizontally more complete picture of the volcanic ash extent" unclear what is meant by more complete picture here – and more complete compared to what?
  We have rewritten the sentence: "Column mass loading observations as

available from, for example, the Spinning Enhanced Visual and InfraRed Imager (SEVIRI) on board Meteosat Second Generation (Schmetz et al., 2002) are beneficial for source inversions as they provide measurements of the horizontal extent of the volcanic ash cloud with a frequency as high as 15 minutes, which is used for analyzing the temporal evolution of the volcanic eruption column. "

- Line 73: Would be good to include some further details of the advantages and limitations of each method which you mention. Also include more recent papers on data assimilation/insertion:
    - Prata, A. T., Mingari, L., Folch, A., Macedonio, G., and Costa, A.: FALL3D-8.0: a computational model for atmospheric transport and deposition of particles, aerosols and radionuclides – Part 2: Model validation, Geosci. Model Dev., 14, 409–436,https://doi.org/10.5194/gmd-14-409-2021, 2021.
    - Pardini, F., Corradini, S., Costa, A., Esposti Ongaro, T., Merucci, L., Neri, A., Stelitano, D., and de' Michieli Vitturi, M.: Ensemble-Based Data Assimilation of Volcanic Ash Clouds from Satellite Observations: Application to the 24 December 2018 Mt. Etna Explosive Eruption, Atmosphere, 11, 359, https://doi.org/10.3390/atmos11040359, 2020.
    - Osores, S., Ruiz, J., Folch, A., and Collini, E.: Volcanic ash forecast using ensemble-based data assimilation: an ensemble transform Kalman filter coupled with the FALL3D-7.2 model (ETKF–FALL3D version 1.0), Geosci. Model Dev., 13, 1–22, https://doi.org/10.5194/gmd-13-1-2020, 2020.
    - Fu, G.; Lin, H.X.; Heemink, A.; Lu, S.; Segers, A.; Velzen, N.V.; Lu, T.; Xu, S. Accelerating volcanic ash data assimilation using a mask-state algorithm based on an ensemble Kalman filter: A case study with the LOTOS-EUROS model (version 1.10), Geosci. Model Dev., 10, 1751–1766, 2017.
    - Fu, G., Prata, F., Lin, H. X., Heemink, A., Segers, A., and Lu, S.: Data assimilation for volcanic ash plumes using a satellite observational operator: a case study on the 2010 Eyjafjallajökull volcanic eruption, Atmos. Chem. Phys., 17, 1187–1205, https://doi.org/10.5194/acp-17-1187-2017, 2017.

Thank you for providing the additional references. We address these now in the introduction. Further, we added some more information to the paragraphs describing the approaches in the literature for analyzing volcanic eruptions. The full section now reads: "First estimations of volcanic ash emissions from the 2010 Eyjafjallajökull eruption in a high temporal and vertical resolution were made by Stohl et al. (2011) and later by Kristiansen et al. (2012) and Kristiansen et al. (2015). Their algorithm bases on the inversion technique of Eckhardt et al. (2008), in which an optimal combination of distinct emission packages is estimated using a least squares method. The method showed to provide reliable a posteriori estimates of the time-varying emission profiles. Stohl et al. (2011) include errors from a priori estimates, retrieval errors and model errors and discussed results in terms of relative error reduction subject to assumptions made. Schmehl et al. (2012) initiate the volcanic ash analysis using an ensemble of simulations with random emission strengths and wind fields. Their best estimate of the volcanic ash concentration is found iteratively using a "genetic algorithm variational approach". Herein, rather strong assumptions on the emission profile are made: the emissions are kept fixed for the simulation duration; emissions are placed into a single model layer; wind fields are only adjusted in the model layer containing volcanic ash emissions. However, the method provides a quick and easy to implement first estimate of the volcanic ash concentrations in the atmosphere. Yet, the strong assumptions may render the approach unfeasible for longer lasting volcanic eruptions in which the

emissions vary more strongly. Another data assimilation method for estimating the volcanic ash emissions was proposed by Lu et al. (2016). They developed an adjoint-free, ensemble-based four-dimensional variational data assimilation (4D-var) method. The method showed reliable estimates of the true emission profile in their experiments using synthetic, vertically integrated satellite observations. However, they do not address the uncertainty estimate of their analysis.

Zidikheri et al. (2016) and later Zidikheri et al. (2017b) developed an assimilation system that aims to analyze the horizontal distribution of volcanic ash column mass loading rather than the emission strength. This study was extended by Zidikheri et al. (2017a) to additionally estimate the height and the particle size distribution of volcanic ash emissions using a parameter refinement method. Here, an ensemble of source parameter values has been applied. Using a proper metric (in their case the pattern correlation coefficient) the ensemble is evaluated against observations. The best fitted ensemble member is taken as analysis. The method is easy to implement for a fast analysis of a volcanic eruption as only the upper and lower bounds of the considered source parameters need to be defined. However, the number of model runs used to find the analysis increases exponentially with the number of parameters. Rough estimates of the parameters' uncertainty are provided by the spread of the top 2 % ensemble members with respect to the metric (Zidikheri et al., 2017b), which does not take uncertainties in the observed quantities into account. Wilkins et al. (2014) used the "data insertion" method, in which observed volcanic ash column mass loadings act as virtual sources for volcanic ash with a predefined vertical distribution. The algorithm was successfully applied to the eruptions of Eyjafjallajökull, Iceland, 2010 (Wilkins et al., 2016b) and Grímsvötn, Iceland, 2011 (Wilkins et al., 2016c). Given the lack of vertical information in column mass loading retrievals of volcanic ash, the data insertion method needs assumptions about the vertical distribution of the volcanic ash content in the atmosphere. Thus, this larger source of uncertainty for the volcanic ash analysis is ignored. The data insertion scheme has also been implemented as a first step towards an ensemble-based data assimilation scheme in the FALL3D-8.0 atmospheric transport model (Prata et al., 2021).

Fu et al. (2017) developed a mask-state algorithm for ensemble Kalman Filters to reduce the size of the state vector to be optimized. More recent applications of the ensemble Kalman Filter and its variants are provided by Pardini et al. (2020) and Osores et al. (2020). By estimating the source parameters of the volcanic eruption, the approaches using the ensemble Kalman Filter assume constant emission parameters between two assimilation steps. This is a rather strong assumption on the emissions especially if observational data is sparse or far away from the volcano. However, keeping this assumption in mind the ensemble Kalman Filter methodology provides an estimate on the analysis uncertainty."

Section 2:

- I generally find the methodology section difficult to follow. There are many technical terms and abbreviations to keep track of, and the descriptions are sometimes not clear. Perhaps an extension of Figure 1 (the flow diagram) to include further steps and references to methods/terminology/naming conventions would help. I also suggest expanding the figure caption of Figure 1 to explain what the figure shows which makes it easier for the reader to return to this figure later while reading subsequent sections.
  Thank you for this suggestion to improve the general description of the method. We have updated the caption of Figure 1 as you suggested. It now reads: "Schematic of the ESIAS-chem analysis workflow. The analysis is initiated with

an ensemble of emission packages at time $t=t_0$ and restarted, when new observations become available (left side, cf. Sect. 2.1). Here, $t_{i+1}$ corresponds to the observation time. Previously calculated simulations with emission packages within the time interval $t_0-t_i$ may be restored (upper panel). Simulated volcanic ash is compared with perturbed observations for the whole simulation (i. e. from $t_0$ to $t_{i+1}$ (upper center panel). The resulting volcanic ash concentrations are passed to the DENM minimization algorithm that produces an ensemble of emission profile analyses (right panel, cf. Sect. 2.2) by finding an optimal combination of the pairwise distinct emission packages. This ensemble of emission profile analyses is evaluated by the particle filter and resampling method to assign a weight to each emission profile according to the fit of the resulting volcanic ash with the observations. Emission profiles are replaced if their corresponding volcanic ash content does not fit well to the observations (lower panel, cf. Sect. 2.3)." Further, we agree that the naming convention of some variable is misleading. We have added a table describing all variables. Also, we have changed the subscripts for referring to ensemble members to superscripts. Thus, ensemble member i is now referred by $(\cdot)^{(i)}$. We are confident that these modifications meet your expectations.

- L 98: "Stohl et al. (2011) and Kristiansen et al. (2015) aiming to estimate the optimal emission profile but not its uncertainty". I think it is fair to say that this work did provide some uncertainty estimates (and included uncertainty in both the a priori, the observations and the model input – though this could of course be improved). In figure 3 of Stohl et al the uncertainty reduction from the a priori via the inversion to the a posteriori is shown. This shows how much influence the observations had, and which parts of the emissions were well constrained by the observations and which were less constrained, and therefore is a form of uncertainty estimate. I do appreciate that what you are aiming to provide is different (but I still don't quite understand how the uncertainty is estimated!) but it would be good to include some more details here how your uncertainty estimate differ from this to make it clear.

Thank you for your comment on the uncertainty. We are aiming to make our point clearer. Our approach accounts for the analysis uncertainty in different ways. First of all, the minimization is performed for an ensemble of emission profiles, where each ensemble member uses perturbed observations and different a priori emission profiles. Further, the ensemble of emission profile analysis is valued by the particle filter algorithm, which assigns weights to each ensemble member and replaces statistically valueless emission profiles, i. e. emission profiles with too little weights. In this way, the ensemble members are comparable in explaining the observed volcanic ash content. The probability is the relative number of ensemble members that simulate volcanic ash concentrations (or column mass loadings) above a threshold. We have added this information to the particle filter section 2.3: "Qualitatively, the strategy of particle filtering applied here can be expressed as follows: By replacing the valueless ensemble members (i. e. those with too little weight) each ensemble member has comparable skill to match the observations. Hence, the probability of an event (e. g. volcanic ash concentrations above a certain threshold) can directly be extracted from the relative number of ensemble members that simulate this event."
Further, Stohl et al. (2011) in their Fig. 3 give the relative reduction of the assumed a priori uncertainty of the ash emissions by the inversion algorithm, with assumptions made and the results further detailed in their section 3.2. We are sorry that we did not appreciate these findings in a pertinent way, what is now made up for. We wrote: "Stohl et al. (2011) include errors from a priori estimates,

retrieval errors and model errors and discussed results in terms of relative error reduction subject to assumptions made." We also add some remarks on our error estimates with particle filter approach. We like to point out that our approach uses the uncertainty estimation to provide reasonable error simulations of the volcanic ash cloud, which has the potential to identify areas with high volcanic ash content without direct observations.

Section 2.2:

- Some further comments on the advantages and disadvantages of the Nelder-Mead method would be good to include. For example, mentioning that the reason the method is suitable for "discontinuous"/"spiky"/"noisy" problems is because it does not use derivatives, but also that it doesn't use a convergence theory and doesn't necessarily find the minimum function value (but rather an 'improvement') – that is the key difference to the method used by Stohl etc.
  We have added some further information on the advantages and disadvantages of the Nelder-Mead algorithm to Sect. 2.2. The section now reads: "The minimization problem posed by (1) is solved using the Nelder-Mead algorithm (Nelder and Mead, 1965). The Nelder-Mead minimization algorithm is a combinatorial optimization method without constraints and without the need to compute the function derivatives. It has proven to be robust, especially in cases where the function to be minimized has discontinuities or the function values are noisy (see McKinnon, 1998). This is expected to be likely in highly variable volcanic eruptions especially given highly uncertainty, and thus noisy, observations. Additionally, the Nelder-Mead algorithm can easily account for bounded regions, in our case positive semi-definite ash loads, and needs relatively few function evaluations (mostly 1-2 function evaluations per iteration, Lagarias et al., 1998).
  The idea of the algorithm is to move a simplex on the surface of the cost function to find an improved model state in a N-dimensional space. The version of the Nelder-Mead method used in this study follows Gao and Han (2012) and utilizes adaptive parameters controlling the step size for each iteration of the minimization. The version has been implemented for parallel operation (Klein and Neira, 2014; Lee and Wiswall, 2007). In our application the Nelder-Mead algorithm is used to find the optimal combination of the pairwise distinct emission packages. Hence, a factor $a_i$, which needs to be scaled by the algorithm, is assigned to each emission package.
  Due to its simplicity the Nelder-Mead algorithm is easy to implement but it is likely to find a local rather than the global minimum of the cost function (which is also a problem for least-square minimization techniques with poor initial guesses, as for volcanic eruptions). Thus, we have added some adjustments to the algorithm. First, we perform the minimization only for integers (including 0). Thus, only integer values are accepted for the scaling factors $a_i$ of the emission packages. By applying this constraint it is assumed that the introduced errors are of lower order than the error introduced by the temporal resolution of the emission packages. Further, the minimization is restarted with larger perturbations of the vertices once the optimization fails for many iterations. Finally, the minimization is started for an ensemble of Nelder-Mead analysis. As perturbed observations are used as input to the minimization procedure, the solutions (here emission profiles) produced by the analysis ensemble are assumed to map the uncertainty given by the observations onto the emission rates. Thus, the minimization

algorithm is called hereafter discrete-grid ensemble Nelder-Mead method (DENM)."

- "The minimization was performed in $N_0^N$, which has been found to be more effective than the minimization in $R^N$" –to help the reader please directly spell out what this means. I think you mean in model space rather than in observation space. And if this is the case is this more efficient because the number of ensemble members is smaller than the number of observations?
Thank you for stating this point to be not clear. We allow only integers as solutions for the minimization. Tests showed that the minimization is less trapped in local minima and the convergence to the solution is faster. We have made this point clearer as can be seen in the response to the previous point.

- How is the initial simplex determined for each ensemble member? You say later it is arbitrary but some more details here would be useful.
Indeed, we start the minimization from an arbitrarily chosen initial simplex (emission profiles in our case). We have tested the algorithm starting from an emission profile with an umbrella shaped vertical mass distribution that varies temporally in strength and plume height. For the given "true" emission profile in our study, we found better performance using the arbitrary initial simplex. It is likely that this is only due to the chosen true emission profile in the nature run. In an application to real volcanic eruptions, we will again test the arbitrary initial emission profile against the time-varying umbrella shaped emission profile. In our algorithm, the initial simplex can be freely chosen, which allows to adapt the method to the characteristics of the current volcanic eruption and its assumed degree of uncertainty. We added this information to section 2.4 where we introduce the arbitrary initial emission profile (we have omitted to use the term simplex to avoid further confusion about the terminology): "The algorithm was tested using a time-varying initial emission profile with umbrella-shaped vertical mass distribution. Due to the chosen true emission profile in this idealized study (cf. Sect. 3) the minimization using the initial emission profile with umbrella-shaped vertical mass distribution shows larger errors. In the application of the algorithm to a real volcanic eruption the performance of the analysis using umbrella-shaped initial emission profile may exceed the performance using arbitrary emission profile. Hence, ESIAS-chem is designed to adjust the initial emission profile to the characteristics of the current volcanic eruption."

- It would also be nice to see some comment on computational effort (i.e. run time) for this system.
As the core focus in this study is the reconstructability of the 3D ash field based on wind shear driven sequences of 2D column field imagery, numerical efficiency was not our primary concern. The run time of the ensemble system is an informative value about the applicability as early warning system. However, as for other methods in the literature, we have decided to not concentrate on the computational performance. Thus, we have adapted the simulations to the available compute resources (especially granted wall clock time). We run the ensemble of emission packages subdivided into chunks of 60 members. Further, we increased the number of iterations in the Nelder-Mead minimization to 15,000 (including restarts), which is not feasible in a realistic early warning scenario. However, we chose 15,000 iterations in order to track the performance of the minimization. It was found that the costs reached the minimum value after ~1,000 iterations. With this setup, the run time of the system is not competitive with other algorithms nor representative for a realistic application.

Section 2.3:

- L 140: "It is noted that in the particle filter method no assumptions of the error statistics of the model state and the observations were made." I don't understand how this relates to the uncertainty estimate you apply in the results section where you assume a 40% uncertainty on the observations….
  We apologize to not have been clear on this point. We mean that the particle filter formulations do not need Gaussian error statistics or unbiased model states as it is necessary in other data assimilation approaches (although generally the ensemble needs to have a large enough spread such that the solution is within the spread). We have added this information to the manuscript: "It is noted that in the particle filter method no assumptions of the statistical forecast error characteristics and the observation error were made (the errors do not need to be normally distributed and the model state does need to be unbiased as other data assimilation methods require)."

L 145 "the ensemble members with high weights are duplicated and perturbed, replacing ensemble members with vanishing weights." I don't quite understand how this works in practice with the unit ensemble members... please expand on this part.
We apologize for the unclear description of the methodology. We have carefully revisited section 2 and changed the wording such that the distinction between the ensemble of emission packages and the analysis ensemble is clearer. We have added some more description to the text: „The ensemble statistics can now be computed using the ensemble member weights. For example, the ensemble mean is

$$\bar{x} = \sum_{i=1}^{N_{ns}} w^{(i)} x^{(i)} "$$

Further, we have added: "In ESIAS-chem, the particle filtering and resampling steps are applied after the ensemble of optimal emission profiles has been found by the DENM algorithm. A weight $w^{(i)}$ is assigned to each optimal emission profile. Residual resampling (Liu and Chen 1998) is used to replace emission profiles leading to too small weights by emission profiles with high weights (this step includes perturbing duplicated emission profiles). After resampling, the weights are normalized again ($w^{(i)}=1/N_{ens}$). Thus, the statistical informative value of the analysis ensemble is preserved.
Qualitatively, the strategy of particle filtering applied here can be expressed as follows: By replacing the valueless ensemble members of the analysis (i. e. those with too little weight) each ensemble member has comparable skill to match the observations. Hence, the probability of an event (e. g. volcanic ash concentrations above a certain threshold) can directly be extracted from the relative number of ensemble members that simulate this event."

Section 2.4

- L 150: "In order to account for meteorological uncertainties, ESIAS-chem is capable to be coupled with ensembles of meteorological field". But you have not used met ensembles here? Please clarify in text if this is only a possible extension.
  Yes, it is a possible extension, which is tested in an upcoming real volcanic eruption investigation. We have changed the sequence as follows: "ESIAS-chem is constructed such that it is applicable to other accidentally released matter and constituents, given constraining observations are available. Further, it is capable to be coupled with ensembles of meteorological fields to account for additional uncertainties resulting from meteorological forecasts. However, this idealized investigation focuses on the ability of the system to reconstruct the emission

profile and its uncertainty under perfect meteorological conditions. Thus, no meteorological ensemble is used."

- L 179: "Finally, a particle filter step is applied. The weights, which result from the filtering step, are applied to the optimized emission profiles." Please provide some more details of how the "ensemble mean" is constructed (a term which you use in Section 3) and then also the uncertainty estimates based on the ensembles. I understand that each ensemble member simulates the emission released by a single emission package for an individual time step and height interval and then weighted by the likelihood in the particle filter step. How do you construct the ensemble mean from this? Are the ensemble members still associated with emissions from a single emission package but after the filtering with an amount of ash (rather than a unit release)?

  Thank you for pointing out that we were not clear in this point. The ensemble of emission packages is input for the Nelder-Mead minimization, which constructs an ensemble of emission profile analyses. These analyses are weighted by the particle filter algorithm. We have added the information on how to calculate the ensemble mean and the probability from the ensemble in Section 2.3 (see also our answer above). Further, we have clarified the input and output of the Nelder-Mead algorithm. To distinguish between the ensemble of emission packages and the ensemble of emission profile analyses (meaning analysis ensemble) we explicitly use these terms instead of only "ensemble" in section 2. In section 3, only the analysis ensemble is referred to. We have carefully revised the manuscript for other occurrences of the word "ensemble", which may cause confusion and changed the terminology accordingly.

Section 3:

- L187: "Given the identical twin assumption the experiment is then to be made realistic in all other respects, as the two different weather conditions on our case". This sentence is not entirely clear. I think you refer to the "twins" as the two different weather conditions. It might be better for the reader if you say this in the first sentence of this paragraph.

  We have added another sentence illustrating the identical twin principle more clearly: "The term identical twin refers to the fact, that observations and a priori knowledge are constructed from the same model and input data, in which only the parameters to be optimized (emission profile in our case) differ."

- L190: Is EURAD-IM an online or offline model? Please clarify and which meteorological data are used as driver/lateral boundary conditions.

  Thank you for this note. Indeed, we have forgotten to give the information about the meteorological model. We have added the following information: "As we consider the differences of feedbacks of the ash clouds on the meteorological evolution as not critical on the forecast time scale in our idealized tests the EURAD-IM is offline coupled with the Weather Research and Forecasting (WRF) model version 3.7 (Skamarock et al., 2008). Meteorological boundary conditions are taken from the ECMWF (European Centre for Medium-Range Weather Forecasts) analysis."

- L203: "The uncertainty of volcanic ash column mass loading observations is about 40 % (Kristiansen et al., 2015, and references therein)": Some better references here might be:
  - L. Clarisse, F. Prata: Chapter 11 - Infrared Sounding of Volcanic Ash Editor(s): Shona Mackie, Katharine Cashman, Hugo Ricketts, Alison Rust,

Matt Watson, Volcanic Ash, Elsevier, 2016, Pages 189-215, ISBN 9780081004050, https://doi.org/10.1016/B978-008-100405-0.00017-3. (https://www.sciencedirect.com/science/article/pii/B9780081004050000173)

- Kylling, A.; Kahnert, M.; Lindqvist, H.; Nousiainen, T. Volcanic ash infrared signature: porous non-spherical ash particle shapes compared to homogeneous spherical ash particles. Atmos. Meas. Tech. 2014, 7, 919–929. 144.
- Western, L.; Watson, I.; Francis, P. Uncertainty in two-channel infrared remote sensing retrievals of a well-characterised volcanic ash cloud. Bull. Volc. 2015, 77, 67.

Thank you for providing further literature. We have surveyed the literature and changed our statement: "The uncertainty of volcanic ash column mass loading observations is about 40% (Western et al., 2015; Clarisse and Prata, 2016) or even higher (Wen and Rose, 1994; Kylling et al., 2014)."

- Equation 7: Here you use an observation error 40% of the observation value. It might be worth commenting here that when using real observations (instead of synthetic as in your case) then a better approach would be to use the retrieval uncertainty estimate for each single satellite pixel, and not a fixed uncertainty value..

We have added this comment to Eq. 7: "For applications to real volcanic eruptions, the observation error provided by the satellite retrieval per pixel should be considered."

- L219+223: "The length of the assimilation window influences the performance of the data assimilation algorithm due to the influence of vertical and horizontal mixing and vertical wind shear." And "Certainly, by increasing the assimilation window length the observations include more information, as the residence time of volcanic ash in the atmosphere is increased". The wording here is a bit strange. The residence time of volcanic ash in the atmosphere doesn't increase with increased assimilation window. As the assimilation window increases the number of observations which are assimilated increases… Please rewrite.

We agree. The wording is insufficient to illustrate our point. The second sentence now reads: „With increased residence time in the atmosphere the volcanic ash at different heights becomes more horizontally split by wind shear. This effect can be exploited by increasing the assimilation window."

- Page 11-12: There is a lot of jumping between the pcc and RMAE. Might be worth explaining first one and the results then the other one and the results.

As was suggested by reviewer 3, we have shifted the definition of the analysis metrics to Section 2. Thus, there is no more jumping between pcc and RMAE and we hope this clarifies the matter.

- L265: 10 μgm−3 seems like a very low concentration threshold (considering the aviation thresholds starting at 200 ug/m3). What was the reasoning behind this threshold, and do the results change if the threshold is higher?

We chose a low threshold to investigate the error of the full volcanic ash cloud, not only the highest ash concentrations. However, the results are the same when choosing, for example, 200 μg/m3 as threshold, although the number of grid cell exceeding this threshold is low, especially in the later hours of the investigation. We have added this information to the text: "The relatively low threshold to calculate the RMAE was chosen in order to increase the number of grid cells to be analyzed and to investigate the full volcanic ash cloud rather than only the area of high concentrations."

- L282: "The analysis suggests that for the respective test cases an assimilation window of 18 hours, that is 10 hours after the artificial eruption terminated, is sufficient for ESIAS-chem to analyze the exact location of the volcanic ash cloud…" This is a nice result (and backed up by the RMAE later) which I think you should put in the abstract.
  Thank you for the suggestion. We have added this result to the abstract: „For our test cases using an artificial volcanic eruption, we found an assimilation window length of 18 hours, i. e. 10 hours after the eruption terminated, to be sufficient for analyzing the extent and location of the artificial ash cloud."
- You could also here refer to work by Fu et al who also analysed this "effective duration" (i.e. the required temporal cycle to obtain improved forecasts). Fu et al. 2015, 2016, 2017 report between 6 and 24 hours.
  - Fu, G.; Lin, H.; Heemink, A.; Segers, A.; Lu, S.; Palsson, T. Assimilating aircraft-based measurements to improve forecast accuracy of volcanic ash transport, Atmos. Environ., 115, 170–184, 2015.
  - Fu, G.; Heemink, A.; Lu, S.; Segers, A.; Weber, K.; Lin, H.X. Model-based aviation advice on distal volcanic ash clouds by assimilating aircraft in situ measurements, Atmos. Chem. Phys., 16, 9189–9200, 2016.
  - Fu, G., Prata, F., Lin, H. X., Heemink, A., Segers, A., and Lu, S.: Data assimilation for volcanic ash plumes using a satellite observational operator: a case study on the 2010 Eyjafjallajökull volcanic eruption, Atmos. Chem. Phys., 17, 1187–1205, https://doi.org/10.5194/acp-17-1187-2017, 2017.
  - Fu, G.; Lin, H.X.; Heemink, A.; Lu, S.; Segers, A.; Velzen, N.V.; Lu, T.; Xu, S. Accelerating volcanic ash data assimilation using a mask-state algorithm based on an ensemble Kalman filter: A case study with the LOTOS-EUROS model (version 1.10), Geosci. Model Dev., 10, 1751–1766, 2017b.

  Thank you for pointing us to the literature. In our study, we state that we need observations at least 10 hours (or later) after the eruption has terminated in order to sufficiently estimate the volcanic ash cloud. The literature provided by you moreover states, that once an improved model state of the volcanic ash is found, the improvement rests for 6 to 24 hours. With our experiment, even though it is an idealized study,  we show that the improved model state prevails until the end of the forecast time. However, we have not focused on the duration of the improvement in the model state due to the assimilation. All this makes it difficult to compare our results with the findings by Fu et al. Thus, we suggest to rather not compare with these publications.

- L305: "good performance of the ESIAS-chem analysis not only in terms of column mass loading but also in terms of the vertical distribution of the volcanic ash in the atmosphere". It might be worth re-iterating here that getting a good performance for the concentrations are possibly strongly affected by the use of "perfect meteorology" and that such good results are not expected using real observations.
  You are right, the perfect model and perfect meteorology assumption contributed to the good performance of ESIAS-chem for the comparison of the vertical distribution of the volcanic ash. This is what we will investigate more clearly when applying ESIAS-chem to real volcanic eruptions, where meteorological forecast uncertainties impose a limiting factor to further improvements. However, the good estimate of the emission profile for the case study with strong wind shear hints that at least for strong wind conditions the vertical distribution of volcanic ash can

be sufficiently analyzed under realistic conditions. We have added the following comment to our statement: "However, this good performance in analyzing the vertical structure of the volcanic ash cloud is partly due to the perfect model / perfect meteorology assumption made in this study. The reliable estimate of the emission profile for the test case with strong wind shear suggest that the vertical structure of the volcanic ash is also sufficiently estimated under real conditions, where meteorological forecast uncertainties impose a limiting factor to further improvements. This needs to be proved in the application to real volcanic eruptions."

- L306: I was surprised why the 18 hours assimilation windows wasn't chosen here over the 24 hours as the 18 h seems to show equally good results up until now. We have chosen 24 hours instead of 18 hours because it is a realistic choice for operational applications. Indeed, we could have shown the 18 hours assimilation window case with equal results.

- You first show the validation using pcc and RMAE (Figs 6-8) for all assimilation time windows, and then the results for one of the assimilation windows (figure 9-10). I would prefer it the other way around so that I can see what the estimated emission profile looks like (for one of the assimilation windows) before it is validated and tested against the other assimilation windows. Also, because the main strength of your method is giving an uncertainty estimate for the emission profile this should be the focus of the results.
  We understand your preference. It is certainly an option to concentrate first on the estimated emission profile for one assimilation window and see how it performs compared to other assimilation windows. Our intention was to first compare the results for the different assimilation windows before concentrating on one realistic choice (as mentioned before, we could have concentrated on the 18 hour assimilation window case, too). As a compromise, we have added the emission profile for the analysis ensemble mean for all assimilation window lengths in Appendix A. Also, we have added an investigation on the error distribution for estimating the emission rates (measured by a relative emission factor between the model and the nature run emissions) for all assimilation window lengths. Both is provided as supplement to this author response.

- Figure 9 and 10: It would be interesting to see "b" figure for all assimilation windows, to see how the emission profile changes as you assimilate more and more observations, and how the estimated uncertainty (c, d figures) also changes when including more observations.
  As mentioned above, we have added this information in Appendix A.

- L322: In the abstract you say that for the strong wind shear condition the estimated emissions have "up to an error of only 10 %" but here you say relative errors are around 10-20 %. Also, in the abstract you say that in a situation with little wind shear the errors are "higher", while here you say up to 60% and more. I would change the abstract to give the same numbers as here.
  Thank you, we have updated the numbers in the abstract.

- L337: "The results indicate that on 29 April 2010 the mixing of volcanic ash in the atmosphere is too effective". With much less wind shear on 29 April it seems that the problem isn't too much mixing but that the emissions at different altitudes and times are transported in a similar way and thus cannot be easily separated by the assimilation.
  Clearly, this is one explanation for the results. However, we believe that our conclusion is correct. As you can see in our Fig. 12, right hand side, the upper and lower part of the volcanic ash cloud is horizontally displaced, which suggests that column mass loading observations should be useful to estimate the emission

profile. The problem here is the long residence time of the volcanic ash cloud above Iceland, which enables the mixing of volcanic ash emitted at different times at the lower part of the emission column. This prohibits a better performance of the system when increasing the assimilation window length as the volcanic ash cannot be attributed to the correct emission package once the volcanic ash has mixed.

- L340: "However, the previous results show that even though the volcanic ash emission profile could not be properly estimated by the system on 29 April 2010, the vertical and horizontal distribution of volcanic ash in the atmosphere is fairly represented by the ensemble mean.". This is a little worrying. The fact that the pcc and RMAE give such good scores even with such a "smooth" emission profile after the assimilation which deviates strongly from the nature run emissions ("truth")... it does make me question whether the pcc and RMAE are appropriate statistical measures to be used here… But it might be more to do with the fact that with little wind shear many different emission profiles can equally well give a best fit with the observations and the fact that the Nelder-Mead method doesn't necessarily find the minimum only an "improvement" as previously mentioned. I think the point that the emissions are not well estimated, but that the simulated concentrations and column loadings still fit well with the nature run would be a little bit more explored and discussed.

  The weak wind shear case, in which the increasing thickness of the nascent ash cannot help to analyze the height-time resolved emission profile, is a typical case of an ill-conditioned inversion problem. Any built-up sequence (upper level first-lower last, or vice versa) in a stagnant column can comply with later observations off the volcano, after inception of wind. The vertical profile of the eruption sequence remains beyond analysability with any inversion method, given only column thickness data. Additional observations, e. g. the Keflavik radar or similar height resolving observation systems, are required to further constraint the volcanic ash emissions. Yet, this is not part of this study.

- L371: "Thus, ESIAS-chem demonstrates to estimate the vertical distribution of volcanic ash in the atmosphere on both simulation days with a high accuracy." A probability of 90% from the ensemble does not mean that the simulation is of high accuracy.

  You are right, the fact that the ensemble predicts an event by 90 % probability does not show its accuracy. As we need a large number of analyses to give reasonable estimates of the accuracy of the method and given the lack of computational time for such a large investigation, we decided to remove this sentence from the manuscript.

Section 4 Conclusions

- This reads a bit more like a discussion and future outlook. I would rename this to "discussion" and include another section for the Conclusions which summarizes the key results you have shown with a few bullet points.

  We have revised the conclusions. Now, Section 4 is "discussion and conclusion". As you suggested, we have added some bullet points to summarize our main findings.

Technical corrections

- Line 33: "form radar" – change to "from radar"
  Done
- Line 285: "extend" – change to "extent"
  Done

We have marked our responses in blue.

The manuscript presents an inversion method (particle filter based) to derive the volcanic ash emission profiles by converting two-dimensional ash loading data from, for e.g., geostationary satellites to three-dimensional emission data. Similar to previous studies the authors combine observations and ensemble simulations. The novel aspect of the method lays mainly in its ability to estimates the errors and uncertainties in the derived emissions. The authors use the inversion system for two notional sub-Plinian eruptions of the Eyjafjallajökull and show that the method's accuracy strongly depends on wind shear conditions.

The methods are valid and the results are interesting for the remote sensing and modeling communities. I have no major comments but few remarks that should be addressed before publication.

We very much appreciate the reviewer comments concerning the readability of our manuscript. We hope this helps to improve the text and to show our findings more clearly. Please see our response below.

General comments:

- The method quantifies the uncertainty with respect to the injection height and vertical wind profile but there is no hint of the uncertainty in the assimilated quantity, i.e. ash mass loading. The satellites do not measure this quantity directly. Rather, it is a retrieved parameter based on brightness temperature. The retrieval has its own limitations and uncertainties. Most importantly, in the first few hours of the eruption the umbrella cloud is quite large and thick so the ash retrievals are either missing or subjected to large uncertainties. Therefore, the authors should discuss the limitations of the method from this perspective.
  The reviewer is certainly right. We did not consider retrieval uncertainties of integrated column values as a possible obstacle to devaluate our concept. The observation error covariance matrix R (defined prior to eq. (3)) accounts for the impact of retrieval errors of the ash column loads and is considered diagonal. It can be made spatially and temporally dependent, to account for assumed increased retrieval errors due to water cloud influences, particularly thick umbrella ash clouds above or in the vicinity of the volcano or interference of other aerosols as e. g. mineral dust. In our study, we have made assumptions about the observation error (including retrieval error). Certainly, in applications to real volcanic eruptions, the use of retrieval errors provided by the observations is highly encouraged.
  Starting from a scalar column load value as exclusive data source we considered estimation uncertainties of the derived height profile presented here as an order of magnitude larger than retrieval errors. We are grateful for the reminder to consider columns sufficiently distant to the nascent umbrella cloud.

The observation error can also be incorporated in constructing the ensemble, as in general any ensemble data assimilation procedure can straightforwardly account for the retrieval uncertainty by artificially perturbing retrievals of column mass loads, where the random perturbation is scaled by the assumed statistics of retrieval errors. This perturbed observation approach is included in our analysis. Clearly, this must not be the only means to generate the ensemble, as this accounts only for a fraction of overall uncertainty, resulting in underdispersive ensembles. We included the remarks above in the formulation of the objectives and the data use description.

- The authors assume that the only parameter relevant for the ash transport is the wind. What about the particle size and aerosol dynamics? How does the method address the uncertainties with respect to processes like particle growth and sedimentation? Ash aggregation leads to particle growth and enhances the removal. Please discuss the limitations of the method from this perspective.
Aerosol dynamics (nucleation, accumulation, deliquescence) and aerosol chemistry in EURAD-IM is based on MADE (Ackermann et al. 1998, with substantial update of the thermodynamical part by Friese and Ebel, 2010, both developed at our research group), which has been switched off for two reasons: Numerical efficiency in an ensemble context and specifics of volcanic ash properties cannot be expected to be reasonably well featured by a general pollutant aerosol module like MADE. Ideally, a full volcanic ash aerosol dynamics and chemistry as proposed by Schmidt, see e.g. https://www.springer.com/gp/book/9783642348389 would be in place, along with its not existing adjoint. Yet we consider the error to be negligible within the evolution time frame addressed in our idealized study. As requested, we added a discussion on the limitations of the method from this perspective in the text.

- The text is very difficult to read. It starts right in the title and then continues with the odd formulation of the first two sentences in abstract. In many places throughout the paper, the verb comes in a passive form at the end of a long sentence. This makes the text very difficult to follow. Please consider writing in an active form and avoid long sentences. Especially sections 1 and 2 contain lots of odd formulations and difficult passages. Section 3 is easier to follow but has some generic formulas related to validation processes. Please move all the formulas and their explanations to section 2.
Thank you for the comments on the readability of the manuscript. We have revised the abstract and the first two sections, with emphasis to avoid long sentences, (and certainly also to follow the fellow reviewers' advices). We agree that the length of the title is poor. GMD guidelines request the model / model version in the title. So there was little space for optimization. Yet we now hope to present some reduced lengths.
As requested, we have moved all formulas related to validation to section 2.

Specific comments:

L1-5 (Abstract): The sentences read odd and are difficult to follow. Please revise.

We have fully revised the abstract to increase the readability of the text. The first sentences now read: "A particle filter based inversion system is presented, which enables to derive time- and altitude-resolved volcanic ash emission fluxes along with its uncertainty. The system assimilates observations of volcanic ash column mass loading as retrieved from geostationary satellites. It aims to estimate the temporally varying emission profile endowed with error margins. In addition, we analyze the dependency of our estimate on wind field characteristics, notably vertical shear, within variable observation intervals."

L25-30: This is not an encouraging opening paragraph. The sentences read odd and are difficult to follow. Besides, there is no clear connection between the points. Please revise

We have revised the first paragraph of the introduction. It now reads: "Emission profiles of volcanic eruptions depend on multiple parameters, such as crater size or exit velocity of the emitted mass. Further, they depend on atmospheric stability and wind profile at the volcano. Many of these parameters are unknown or difficult to measure exactly. This renders the estimation of emission profiles of volcanic eruptions challenging for chemistry transport models in the context of data assimilation and inverse modelling for source estimation. Therefore, special methods for assessing the strength and vertical distribution of volcanic emissions are necessary. As volcanic eruptions contain enormous amounts of harmful trace gases and particulate matter, a detailed knowledge not only about the spatial and temporal variations of the emissions and its strength is needed but also accurate information about the analysis error of the emissions and the evolving volcanic ash cloud is required."

L38: please add the specific uncertainties of these methods. Besides, add a

We have added the description of uncertainties of the methods to the text: "Statistical models base on observational data from historic volcanic eruptions, which are sparse and show a large variance in eruption rate for given plume heights. For example, Mastin et al. (2009) calculated an uncertainty by a factor of four in estimating the emission rate for a plume height of 25 km using their statistical model. Physical plume-scale models require orographic details of the volcano (e. g. crater size) but also meteorological fields and parameters (e. g. wind entrainment coefficients), which are often poorly known and render these models highly uncertain. Costa et al. (2016) identified the wind entrainment coefficient as main source of uncertainty leading to up to two orders of magnitude differences for the estimation of mass eruption rates for weak volcanic eruptions. In their analysis of the eruptions of the Eyjafjallajökull, Iceland, in 2010 and Grímsvötn, Iceland, in 2011, Woodhouse et al. (2015) found a comparable range of uncertainty depending on the choice of the wind entrainment coefficients."

L108: you mean "It should be noted" or "we note"?
We have change "it is noticed" to "Please note".

Figure 2: I do know that this is an idealized set-up. But is it physically realistic to have the same profiles and emission rates under two different atmospheric conditions (wind shear)?

Vertical wind shear and associated horizontal wind conditions are found to make the main difference of ash cloud analysability, given only estimated column mass loadings. We have chosen to analyze a special case of volcanic eruption with two short but strong emission pulses in a row. We chose this emission profile to test the method's ability to distinguish the two pulses under different wind conditions. Indeed, if we apply the same eruption parameters to a plume model we would expect differences in the emission profile and rates. However, the emission profile used in our analysis remains realistic to occur, even though differences in the source parameters (exit velocity, water content, etc.) could be expected.

Figure 4 and 5: I did not find the source of meteorological data in the text.
We have added the information about the meteorological data in the text: "As we consider the differences of feedbacks of the ash clouds on the meteorological evolution as not critical on the forecast time scale in our idealized tests, the EURAD-IM is offline coupled with the Weather Research and Forecasting (WRF) model version 3.7 (Skamarock et al., 2008). Meteorological boundary conditions are taken from the ECMWF (European Centre for Medium-Range Weather Forecasts) analysis."

Figure 8: what happens at 12 hours after the eruption. These is an spike in the error.
Figure 8 shows the relative mean absolute error of the (unobserved) volcanic ash concentrations. Thus, the volcanic ash concentration of the ensemble mean is compared to the nature run, from which observations have been extracted, in each grid cell. At 12 UTC, the spike in the RMAE results from the error in estimating the emission strength in the upper part of the eruption column. This error is smoothed out in the subsequent hours reducing the RMAE again. In Appendix A shows the emission profile for each assimilation window. Here you can see the errors of the test case using a 12 hour assimilation window in estimating the second eruption plume around 7 UTC, which leads to the large errors. We have added this information to the discussion of the figure.

Conclusion: Again very hard to follow. Please make it clear and concise.
We have revised the full conclusion to make it clearer.

Literature:

Ackermann, I. J., Hass, H., Memmesheimer, M., Ebel, A., Binkowski, F., Shankar, U.: Modal aerosol dynamics model for Europe: development and first applications, Atmospheric Environment, 32, 2981-2999, 1998.

Costa, A., Suzuki, Y. J., Cerminara, M., Devenish, B., Ongaro, T. E., Herzog, M., Van Eaton, A., Denby, L., Bursik, M., de' Michieli Vitturi, M., Engwell, S., Neri, A., Barsotti, S., Folch, A., Macedonio, G., Girault, F., Carazzo, G., Tait, S., Kaminski, E., Mastin, L., Woodhouse, M., Phillips, J., Hogg, A., Degruyter, W., and Bonadonna, C.: Results of the eruptive column model inter-comparison study, Journal of Volcanology and Geothermal Research, 326, 2–25, 2016.

Friese, E. and Ebel, A.: Temperature Dependent Thermodynamic Model of the System H+−NH4+−Na+−SO42−−NO3−−Cl−−H2O, J. Phys. Chem. A, 114 (43, 11595-11631, 2010.

Mastin, L. G., Guffanti, M., Servranckx, R., Webley, P., Barsotti, S., Dean, K., Durant, A., Ewert, J. W., Neri, A., Rose, W. I., Schneider, D., Siebert, L., Stunder, B., Swanson, G., Tupper, A., Volentik, A., and Waythomas, C. F.: A multidisciplinary effort to assign realistic source parameters to models of volcanic ash–cloud transport and dispersion during eruptions, J. Volc. Geotherm. Res., 186, 10–21, 2009.

Skamarock, W. C., Klemp, J. B., Dudhia, J., Gill, D. O., Barker, D. M., Duda, M. G., Huang, X.-Y., Wang, W., and Powers, J. G.: A description of the advanced research WRF version 3, Tech. rep., NCAR Technical note NCAR/TN-475+STR, 2008.

Woodhouse, M. J., Hogg, A. J., Phillips, J. C., and Rougier, J. C.: Uncertainty analysis of a model of wind-blown volcanic plumes, Bull. Volcanol., 77, 83, https://doi.org/10.1007/s00445-015-0959-2, 2015.

We have marked our responses in blue.

The authors present an idealised study of a particle-filter-based inversion system to obtain height- and time-resolved volcanic ash emission estimates using ash column loads from satellite. The idealised study provides a good testing ground for their inversion method, without the complication of modelling errors and incomplete observation datasets which are encountered in real situations. I think it is suitable for publication but could benefit from some improvements to the readability and complexity of the manuscript. It is an interesting study and I'm keen to see future developments on this work.

We thank the reviewer for the encouraging comments and detailed suggestions helping to improve the manuscript. We give our responses below.

- I find the manuscript unnecessarily wordy in places and not the easiest to read. For example, the title is rather detailed - is 'sub-Plinian Eyjafjallajokull' and 'version 1.0' necessary here. Another example is the caption of Figure 2 – stating it shows the emission profile should be adequate – I would question the need to include what it is to be used for here. The authors use some complex words (e.g., 'pairwise distinct' (line 91) – does this mean independent? – 'investigated exemplary' (line 306) – what does this mean?). In places, sentences are very long. Section 3 is rather long – could it be split into subsections? Readability could be improved, I feel.

  Thank you very much for this helpful comment. We agree that the title is long. However, we have had discussed the title with the technical editor and came to the conclusion that the title should contain information about the eruption type („hypothetical sub-Plinian Eyjafjallajökull eruption") as well as the method characteristics („particle filter based"). Further, the naming of the model and model version is a requirement by GMD and cannot be removed. Unfortunately, this prerequisite almost doubles the length of the title. However, we have removed "the chemical component of" from the title and hope this improves the readability of the title.

  We agree that the caption of Fig. 2 contains unnecessary information. We have changed the caption to: „Hovmoeller-like plot of the nature run emission profile used in this study. Shown is the emission rate (colored) for a given time (x-axis) and height above the volcano (y-axis)."

  The mathematical phrase "pairwise distinct" implicitly means that no members of a set are equal to another. Here the phrase refers to the fact that every emission package, which is defined to be pairwise distinct, covers a unique time and height spot in the emission profile. We have added this information to the phrase: "The simulation is realized by an ensemble, in which each ensemble member simulates the dispersion of one single emission package. Thus, each emission package covers a unique time and height spot in the emission profile. We refer to this ensemble as ensemble of emission packages."

We have removed the phrase "investigated exemplary". The respective sentence now reads: "As an example, the analysis results using an assimilation window of 24 hours are investigated in more detail."

We have revisited the full manuscript with special focus on long and complicated sentences as was also suggested by the other reviewers. Further, we have split Sect. 3 into subsections.

- It would help to have some connection between the theory (section 2) and the practice for this case study.

  We have added a table explaining the variables used in this manuscript. We have also carefully revisited Section 2.4. Here, we have added information about how the system is applied to volcanic ash eruptions and how the theory is used in practice.

  - Line 102: c is defined as a default mass of ash in the emission package but just previously on line 92 'a unit mass of ash' is stated. I began to wonder why c was needed if it was one. Reading on, I realised that c is probably a first guess (e.g. from a prior) and that a is some adjustment factor. This could be made clearer. Again, on line 153 there is reference to 'default mass' without this ever being defined.

    Indeed, we have not been clear here. c is a default mass of ash, which can be arbitrarily chosen. It defines the resolution of the emission strength in our estimate. We have made this point clearer in the manuscript. The variable c is now consistently defined as "default mass of ash" throughout the manuscript, which is scaled by the factor **a**. This factor **a** is to be optimized in the DENM algorithm.

  - The use of the Nelder-Mead algorithm seems unnecessary since the problem here is linear and the resulting minimization problem is quadratic. The situation described on lines 114 and 115 ('cases where the function to be minimized has discontinuities or the function values are noisy') does not apply here. Again, this seems to overcomplicate the manuscript and is an example of the gap between the theory stated and the case study. Furthermore, the required restriction to only allow integer values for a for the method to be efficient (I presume the use of the word 'effective'is a typo?) is unnecessary for such a simple minimization problem which could be solved explicitly and will no doubt introduce some uncertainty / errors. I take the authors point that the Nelder-Mean algorithm is more widely applicable but wider applications are not studied.

    Basically, we fully agree with the reviewer that the minimization problem is quadratic, apart from the bounding due to positive semi-definiteness of all components (i.e. a bounded minimization problem). In fact, we started our study with quasi-Newton L-BFGS, where we always have had best success compared to other methods like CG (since the beginning of our chemistry data assimilation activities with 4D-var, Elbern and Schmidt, 1999). (The L-BFGS-B(ounded) version is very inefficient for parallelization, due to a Gauss-Seidel type problem.) To our experience, essential to the success of high dimensional quadratic minimization problems are two items in data assimilation: introduction of a background

state reasonably close to the "truth" for a tangent-linear approximation to hold, (traditionally provided by a preceding forecast), and, to some extent linked with that, an efficient preconditioning (see e.g. Elbern et al., 2007, for a preconditioning technique by diffusion approach (Weaver and Courtier, 2001, in atmospheric science). With an increasing number of model levels and their (positive semi-definite) concentrations to be attributed, while column values as given data are single scalars only, the ill-conditioning of the minimization problem increases drastically and a much needed reasonable background information prior to the volcanic eruption is hardly available. Hence, this missing a priori knowledge cannot serve any preconditioning requirements other than highly speculative inferences from assumed eruption type and strength scenarios. Even simple smoothness assumptions of the vertical profile are often invalid for ash clouds. While we presently test to maintain positive semi-definiteness by assuming log-normal distribution underlying the least square minimization, we performed tests with the Nelder-Mead method, which performed clearly best, without getting lost in drastically elongated minima as introduced by underdetermined degrees of freedom through vertical level concentrations. Yet we agree that after an initial coarse Nelder-Mead minimization, a final least square step will have the potential to prove successful in the future. We add a modified discussion to this issue in the text.

- Line 133: What is 'the model state'? Presumably it is either the source emissions (a times s used earlier) or the model predictions of ash column loads (both H M[as] and x hat (equation 2) used earlier). Can you standardise the notation? Is x in equation 3 related to x hat in equation 2? If not, perhaps another variable other than x could be used for one? Also, whilst notation is consistent for x and y, the subscripts in equation 8 suggest to me derivatives or (x,y) coordinate components.

  We have added a table that explains the used variables. We have further extended equation 2 to show the link between M(as) and x. Equation 8 does not show the derivative but the deviation from the average. The subscripts do not refer to (x,y) coordinate components but to the modeled and observed volcanic ash detection. The notation using the prime symbol " ' " was introduced in Zidikheri et al. (2016). Thus, we suggest to keep the notation in Equation 8 as is.

- Lines 150 – 151. Does the mention of the capability of using ensemble meteorological members add anything to this paper? This capability isn't used, and its mention could cause confusion to the reader – what does 'ensemble' refer to hereinafter? Lines 191 - 192: 'EURAD-IM comes with the adjoint code of the chemical and aerosol modules for four–dimensional variational data assimilation.' – again, is this relevant to this study?

  We agree that our mentioning of the meteorological ensemble may cause confusion. However, it is an important feature of ESIAS-chem, which is used in subsequent analyses. Thus, we have rewritten the sentence: "Further, it is capable to be coupled with ensembles of meteorological

fields to account for additional uncertainties resulting from meteorological forecasts. However, this investigation focuses on the ability of the system to reconstruct the emission profile and its uncertainty under perfect meteorological conditions. Thus, no meteorological ensemble is used at this stage."
We have removed the sentence concerning the adjoint of the EURAD-IM to avoid further confusion.

- Detail on the error covariance matrix B is thin. On one hand you say, 'no fixed assumptions have been made for matrix B' but later you state that 'B is chosen as diagonal'. How is the optimal value found?
  We agree, we have discussed the B matrix insufficiently. We have added more detail to the estimate of the B matrix. Our point is, that the matrix B can be altered to include constraining information (e. g. correlations of different emission packages). In this study, for simplicity and lack of knowledge on height-time eruption parameters we have decided to not constrain the emission packages, leaving matrix B to be diagonal. The actual values were found by sensitivity runs. Thus, the sequence now reads: "In first tests without the regularization term, the emission rates have partly increased to unrealistic high values. Therefore, the B-matrix was chosen in a sequence of sensitivity tests, in which the influence of the regularization term on the emission profile was evaluated. Best results have been found by choosing B as diagonal matrix B=diag(10). Please note that the chosen diagonal form of the B-matrix led to reasonable results for the artificial emission profile used in this study. However, for realistic applications a more elaborated evaluation of a properly chosen B-matrix is required and straightforwardly applicable. In this performance test, the only purpose of the matrix serves to restrict the scaling factors **a** not to vary too strongly. In addition, the regularization term was chosen in order to maintain a suitable spread of the analysis ensemble."
- Presumably the subscript 0 in line 169 is an iteration subscript?
  Yes, the subscript 0 indicates the first guess. However, we have removed the formula because it adds no further information to the manuscript and only causes confusion. We have changed the related sentence to make our point clearer: "The minimization is initialized with a set of arbitrarily varying scaling factors **a** for the pairwise distinct emission packages."
- It would help for sections 2.3 and 2.4 to be linked better. For example, how does the weightings and likelihood relate to the cost function?
  We have carefully revised section 2.4 to better link the theory of the different methods.
- I'm presuming the 'ensemble mean' (line 242) is obtained from the ensemble members that are accepted by the particle filter, weighted according to the weights in equation 5? It would help the reader, to elaborate here. Also, I find the subsequent mention of 'mean' (line 249) and the 'mean' and the overbar (line 251 and equation 9) confusing. Also, on lines 263 – 264, the ensemble mean is denoted by an overbar. Can any improvements be made to help the reader negotiate these apparent

different means?

We have added some information on how to calculate the mean in our analysis. As the ensemble members are resampled, the weights are not applied by calculating the ensemble mean. We have made clearer, that **va** is the binary volcanic ash detection vector for the analysis ensemble mean and that the overbar in line 49 denotes the spatial average of the volcanic ash detection vector **va**. We hope that this avoids further confusion.

- I would have liked the authors to state early on (perhaps in the abstract) that the study is an idealised study (I think the word 'idealised' is more commonly understood than 'identical twin') and therefore does not consider errors in the modelling (both in the input meteorology and in the model parametrizations) nor incomplete observations. I would also have liked to have seen more discussion of the implications of this work to real life situations (e.g., when errors will exist in the meteorological data and in the transport model and observations may be incomplete (perhaps due to the presence of meteorological cloud)). For example, the authors state that 'an assimilation window of 24 hours is sufficient in order to provide reliable forecasts', but this may not be true for a real case study.

In the revised abstract, we have added that we demonstrate the system's validation in an idealized setup. Further, by introducing the term identical twin experiment we add this information again. We have also changed the headline of Section 3 from "Identical twin experiments" to "Validation of ESIAS-chem". We suggest leaving the term identical twin experiment as it refers to a special case of idealized studies, well defined in the data assimilation community (e.g. Daley 1991). We believe that our updated explanation of identical twin experiment is sufficient to avoid further confusion.

We have revised the discussion and conclusion section and have added a discussion about the limitation of the current study to generalize the results.

- I suggest some thought is given to the use of the phrase 'data assimilation'. What is meant by the term 'data assimilation'? Some uses of the phrase I would refer to as source inversion methods, rather than data assimilation.

Thank you for your suggestion. We agree that the term data assimilation is not precise enough. We used the term data assimilation to acknowledge the use of observations and model simulations to provide improved model analyses. Indeed, our main goal is the source inversion for volcanic ash emissions. We have revisited the manuscript and changed "data assimilation" to "source inversion" were appropriate.

- Lines 253 – 254: Why is the RMAE calculated over points where both the modelled and observed values are above the limit? If one was to compare model forecasts for a given set of observations, the RMAE may be obtained over a different set / number of (model, obs) pairs. If, however, one was to compare the RMAE calculate over points where the observed values are above a limit, one would have a consistent set. Why is the relative error (equation 11) normalised by the model ensemble mean (similarly equation 12)? The RMAE is normalised by the observations.

We thank the reviewer for this note. Indeed, the RMAE was calculated over different datasets. Our intention was to exclude low volcanic ash values from the

comparison. However, this is also accomplished by taking all points, where the observation is above the threshold. We have changed the figure accordingly. The results are similar to the previous one.

We have chosen to normalize the relative error in Equation 11 by the modeled ensemble mean to be comparable by the relative standard deviation, for which the normalization with the model ensemble mean is more intuitive. To be consistent with the RMAE, we have normalized the relative error and relative standard deviation by the nature run emissions. As the maximum emissions of the nature run and analysis ensemble mean are comparable, the results remain valid.

- I would encourage the authors to add some model runtime information.

This has also been suggested by Nina Kristiansen (reviewer 1). As the core focus in this study is the reconstructability of the 3D ash field based on wind shear driven sequences of 2D column field imagery, numerical efficiency was not our primary concern. The run time of the ensemble system is an informative value about the applicability as early warning system. However, as for other methods in the literature, we have decided not to concentrate on the computational performance. Thus, we have adapted the simulations to the available compute resources (especially granted wall clock time). We run the ensemble of emission packages subdivided into chunks of 60. Further, we have increased the number of iterations in the DENM minimization to 15,000 (including restarts), which is not feasible in a realistic early warning scenario. However, we chose 15,000 iterations in order to track the performance of the minimization. We found that the costs reached the minimum value after ~1,000 iterations. With this setup, the run time of the system is not competitive with other algorithms.

Minor points

- Lines 140-141: 'no assumptions of the error statistics of the model state and the observations were made'. Is this true? The likelihood function commonly includes an error covariance term (as in equation 1 and equation 6), hence I would think there are some assumptions made in calculating the weights, even if the error covariance term is the identity matrix.

We see that we have not been clear on this point. The particle filter methodology is applicable to all kinds of probability density functions and is not restricted to Gaussian model and observation errors. We have added this to the text: "It is noted that in the particle filter method no assumptions of the statistical forecast error characteristics of the model state and the observation error were made (the errors do not need to be normally distributed and the model state does not need to be unbiased as other data assimilation methods require)."

- Line 26: 'Chemistry transport models have limits in estimating the emission strength'. The emission strength is usually assumed in chemistry transport models. Models are not generally used to estimate emission strengths, except in the context of data assimilation / source estimation methods or by some simple inference from observations.

You are right. We have changed this statement in the revised introduction. This

first paragraph now reads: "Emission profiles of volcanic eruptions depend on multiple parameters, such as crater size or exit velocity of the emitted mass. Further, they depend on atmospheric stability and wind profile at the volcano. Many of these parameters are unknown or difficult to measure exactly. This renders the estimation of emission profiles of volcanic eruptions challenging for chemistry transport models in the context of data assimilation and inverse modelling for source estimation."

- Line 31: 'analysis error' – it's not clear whether this refers to the emission estimates or the predicted cloud.
  In principle, this statement is valid for both, the emissions estimate and the predicted volcanic ash cloud. We appended "of the emissions and the volcanic ash cloud" after "analysis error" to better illustrate this.
- Line 37: 'and thus making' should be 'and thus make' or 'thus making' or something similar.
  We have removed the full sentence in the revised introduction.
- Line 44: Satellite retrieval methods also usually retrieve an estimate of the cloud height.
  Thank you very much for pointing out that we need to be clearer at this point. We agree that there are many retrieval methods, exploiting infrared satellite measurements to obtain volcanic ash properties including retrieved cloud height information (e.g. Ventress et al., 2016 or Piontek et al., 2021). However, these only include cloud top height retrievals and give no information on the vertical extent of the ash cloud or the averaging kernel. In the sentence (line 44), we referred to data sets as used in Stohl et al. (2011) and Prata and Prata (2012). In these studies, an ash cloud height (and ash cloud thickness) has been roughly estimated according to the observed brightness temperatures. This cloud height then serves as retrieval input and is not included in the retrieved data sets. We now adjusted the related sentence to: "In contrast to lidar observations […], column mass loading observations rarely provide information about the vertical distribution of volcanic ash and are mostly limited to cloud top heights (e.g. Ventress et al., 2016 or Piontek et al., 2021)."
    - Line 52: 'remains to be solved'. I would probably dispute this. Established source inversion methods for volcanic ash use atmospheric wind shear to be able to determine the three-dimensional ash cloud information from two-dimensional observations.
      You are right. We give some examples of these inversion methods in the subsequent paragraph. Thus, we have removed this statement from the manuscript.
- Line 73: 'but also may the'. Something is wrong here with the English – perhaps remove 'may'?
  We have changed the sentence to: "They found that not only the ensemble statistics should be evaluated but also the single ensemble members, which may contribute significant information to the distribution of volcanic ash."
- Lines 96-98: The work of Stohl et al and Kristiansen et al estimates the source emission profile (from which the volcanic ash column mass loading can be modelled) and the Bayesian method used does provide uncertainty information.

Indeed, we are sorry for not have acknowledged their work in a pertinent way. We have added this information to the text. In addition to Stohl et al. (2011), in our approach the uncertainty estimation of the emission profile is used provide probabilistic estimates of the volcanic ash cloud extent. This has the potential to identify areas with high volcanic ash content that are not directly observed.

- Line 150: 'suitably' should be 'when suitably'?
  We have change "suitably" into "given"
- Line 187: 'on our case' should be 'in our case'?
  We have removed this half sentence in the revised version of the manuscript.
- Line 198: 'Spin' should be 'Spinning'
  Done
- Lines 223-224: 'Contrary, vertical and horizontal mixing of volcanic ash emitted may limit the benefit that is gained by increasing the assimilation window length.' I can't see why this would be the case – can the authors explain? I can see that additional observations from increasing the assimilation window length may not provide any further information (particularly in an idealised study) but why the reference to 'vertical and horizontal mixing'? Is it because the ash cloud may be below satellite detection limits when widely dispersed? Does the benefit differ depending on whether the study is an idealised study (i.e., no modelling errors and full view of the ash cloud) or a real case study (with modelling errors and missing observations)?
  This statement refers to the distinction of emission packages given vertically integrated ash column data. Once the volcanic ash emitted by different emission packages is well mixed due to vertical or horizontal mixing, it is impossible to attribute the volcanic ash to one or the other emission package. This effect is independent of our idealization in this study. We have added this example to the text: "For example, if volcanic ash emitted by two different emission packages is mixed, it is impossible to attribute the volcanic ash to one or the other emission package."
- Caption Figure 4: 'in approx. 5 km' should be 'at approx. 5 km'. This typo appears in a few other places in the manuscript (e.g., lines 326, 355, 396) – it should be 'at' a height, not 'in' a height.
  Thank you for this note. We have changed this typo throughout the manuscript.
- Line 229, Figure 4b. It's not clear whether this is height above the volcano (which probably doesn't make sense over a spatial region) or height about ground or height above sea level? How does one associate it with Figure 4a? Does one need to know the height of the volcano? Similarly, line 231. Lines 231 – 232 and Fig 4c require some units / labels for the variables stated / shown (e.g., label on the y axis, units for temperature and pressure). Similarly, for Figure 5 and associated text. Also, height information is not specific in Figures 9-12 and associated discussion.
  Thank you for mentioning this issue. The wind speed in Fig. 4b is actually plotted on pressure level 500 hPa and not at a height of 5 km. We apologize for the typo. We have changed the caption of Figs. 4 and 5: "Meteorological conditions on 15 April 2010. (a) Wind speed above the volcano for the whole simulation period. (b) Wind speed at 500 hPa on 15 April 2010, 12 UTC, which corresponds to approx.

5 km above the volcano. (c) Vertical cross-section of isobars in [hPa] (red) and isotherms in [K] (grey) along the red line in b) on 15 April 2010, 12 UTC." We have changed the text accordingly.
We added a label to the y-axis in Figure 4c. The units of isobars and isotherms are added to the caption. Similar changes have been made in Figure 5 and its caption.
In Figs. 9 and 10, the height is referred to the height above the volcano. Also, in Fig. 11 and 12, the height is referred to the height above the ground. We have changed the label of the y-axis accordingly.

- Lines 279-280: 'Again, the pattern correlation coefficient does not account for deviations in the strength of volcanic ash column mass loading at locations in which the ensemble mean and the nature run differ in volcanic ash load'. It considers differences above and below the limit applied.

  You are right. We have changed the sentence to: "However, the pattern correlation coefficient is a measure for volcanic ash column mass loading above and below the chosen threshold. It does not measure differences in the strength of volcanic ash column mass loading above the threshold."

- Line 316: Given the same emission profile is used in each nature runs, why are these total values different?

  Actually, the emission profiles slightly differ, which is not visible with the chosen colorbar. This difference results from the calculation of the emission profile in the underlying EURAD-IM model, in which the model layer depth is taken into account.

- Line 337: 'the mixing of volcanic ash in the atmosphere is too effective'. This study is an idealised case study so the mixing of volcanic ash in the atmosphere is represented perfectly. My opinion is that the second case study does not enable the vertical distribution of the ash emissions to be determined by wind shear and hence the filtering method yields an emission profile which is widely distributed in the vertical compared to the nature run. The first case study has significant wind shear which allows the vertical distribution of emissions to be strongly constrained, but this is not possible for the second case study.

  We understand your interpretation of our results. The lack of vertical wind shear is one limitation for the estimation of the emission profile for the second test study. However, our results may also suggest alternative causes, which we would like to discuss briefly: Fig. 10b shows the mean emission profile of the analysis ensemble. Especially for the first eruption column around 3 UTC, the emission rates are underestimated in the full vertical column. Thus, the error is unlikely due to the vertical distribution of the emission rates. Compared to the nature run, lower emission rates in the eruption column are rather compensated by strong emissions at later hours. This leads to a temporally highly smoothed emission profile. In addition, Fig. 8b shows a low relative mean absolute error of the volcanic ash concentrations. Thus, we conclude that the dispersed volcanic ash cloud resulting from the temporally smoothed emission profile for the second test case is similar to the volcanic ash cloud resulting from the nature run's emission profile.

- Line 381: It's not just the uncertainties in the meteorological fields which are neglected in this study, uncertainties in the model parametrizations (e.g., turbulent dispersion, washout, etc.) are also neglected.
  You are right. We have added this to the discussion section: "The analysis is idealized in different ways: The uncertainties in meteorological fields, especially in winds, in model parameters (e. g. deposition velocity), and parametrizations (e. g. clouds) have been neglected. Further, the amount of observational data is exceptionally large, with observations of the full domain every 6 hours. Thus, observations of ash-free areas allow for removing volcanic ash emissions from the analysis. The ability of ESIAS-chem to give reliable results for real volcanic eruption using non-idealized meteorology and incomplete observations needs to be addressed in another study."

Literature:

Daley, R.: Atmospheric Data Analysis, Cambridge Univ. Press, 1991.

Elbern, H. and Schmidt, H.: A four-dimensional variational chemistry date assimilation scheme for Eulerian chemistry transport modeling, *J. Geophys. Res.*, 104, 18583-18598, 1999.

Elbern, H., Strunk, A., Schmidt, H., and Talagrand, O.: Emission rate and chemical state estimation by 4-dimensional variational inversion,Atmos. Chem. Phys., 7, 1–59, 2007.

Piontek, D., Bugliaro, L., Kar, J., Schumann, U., Marenco, F., Plu, M., and Voigt, C: The New Volcanic Ash Satellite Retrieval VACOS Using MSG/SEVIRI and Artificial Neural Networks: 2. Validation. *Remote Sens.* **2021**, *13*, 3128. https://doi.org/10.3390/rs13163128

Prata, A. J., and Prata, A. T. (2012), Eyjafjallajökull volcanic ash concentrations determined using Spin Enhanced Visible and Infrared Imager measurements, *J. Geophys. Res.*, 117, D00U23, doi:10.1029/2011JD016800.

Stohl, A., Prata, A. J., Eckhardt, S., Clarisse, L., Durant, A., Henne, S., Kristiansen, N. I., Minikin, A., Schumann, U., Seibert, P., Stebel, K., Thomas, H. E., Thorsteinsson, T., Tørseth, K., and Weinzierl, B.: Determination of time- and height-resolved volcanic ash emissions and their use for quantitative ash dispersion modeling: the 2010 Eyjafjallajökull eruption, Atmos. Chem. Phys., 11, 4333–4351, https://doi.org/10.5194/acp-11-4333-2011, 2011.

Ventress, L. J., McGarragh, G., Carboni, E., Smith, A. J., and Grainger, R. G.: Retrieval of ash properties from IASI measurements, Atmos. Meas. Tech., 9, 5407–5422, https://doi.org/10.5194/amt-9-5407-2016, 2016.

Weaver, A. and Courtier, P,: Correlation modelling on the sphere using a generalized diffusion equation, Q. J. R. Meteorol. Soc., 127, 1815-1846, 2001.

Zidikheri, M. J., Potts, R. J., and Lucas, C.: A probabilistic inverse method for volcanic ash dispersion modelling, in: Proceedings of the17th Biennial Computational

Techniques and Applications Conference, CTAC-2014, edited by Sharples, J. and Bunder, J., vol. 56, pp.C194–C209, 2016.

---

## Editor Decision (ED1)

Topical Editor Decision for "Vertically-resolved probabilistic volcanic ash analysis using the chemical part of the Ensemble for Stochastic Integration of Atmospheric Simulations (ESIAS-chem) version 1.0" by P. Franke, A. C.. Lange and H. Elbern.

I completed two initial Topical Editor reviews of this manuscript (beginning and end of April 2021), a version with revised Abstract and Introduction proceeding then to expert peer review in May 2021.

Three reviewers posted comments during the review period (May to July), with the authors uploading a revised manuscript, ATC version and replies to the reviewers comments at the end of September.

I can see that the authors have replied comprehensively to each of the reviewers' comments, in each case their comments minor in nature, and the manuscript has improved substantially as a result.

All three expert reviewers recommended publication once the required revisions were made, and I can confirm that the authors have addressed these appropriately, both in their replies and in the revised manuscript.

Upon reviewing the revised manuscript however, I did notice two areas where I still feel some aspects are not sufficiently explained.

I am therefore recommending publication once 4 further issues are remedied, or explained, this additional Topical Editor review only for these few minor comments however.

1) On page 13 (lines 340-341), the authors explain:

"Column mass loading of volcanic ash in [$gm^{-2}$] is extracted as fictional observation data $y_i$ , every 6 hours, from a 'nature run', simulated by the forward model of EURAD-IM. "

I understand the basis for this approach, in using the higher resolution EURAD-IM dispersion model, for two alternative scenarios of a sub-Plinian Eyjafjallajokull eruption, to provide synthetic observations that can then be used to test the ESIAS-chem simulations with the the data assimilation.

What I don't understand though, is that the text states (line 340) that these "fictional observation data" are extracted from the model only every 6 hours, this being presented then as a proxy for a data stream representative of the SEVIRI geostationary satellite.

Clearly the geostationary satellite will provide measurements at a much higher temporal resolution than "every 6 hours", with data every 30 mins or 1 hour being how the system will then be able to adjust/weight its ensemble predictions with the technique described.

I'm assuming that "every 6 hours" must be the authors referring to the data-flows, with perhaps a block of 12 sets of 30-min resolution data extracted every 6 hours.

That might well have been obvious to the expert reviewers, but as currently worded, that's a confusing mis-match to the approach for the synthetic observations representative of a geostationary satellite's monitoring of the volcanic plume/cloud's progressing dispersion.

Related to this point, reading the reviewer's reply to the 1st of reviewer 1's comments, to revised the Abstract sentence previously beginning "The system validation", now revised to "Thus, the proposed system" (lines 5-6), I'm suggesting also to highlight the benefits of the geostationary datasets, that primarily being the high temporal resolution that can then constrain how the model predicts that the plume/cloud develops in the initial days (e.g. with the ash particles sedimenting alongside any co-emitted sulphur dioxide oxidising to sulphate aerosol, and the evolving wind shear etc.).

I'm not suggesting to add those specifics of the particles involved, but simply to add "2D high temporal resolution" before "column mass loading data" within that new merged sentence of the Abstract. Also, the word "imagery" (at the end of that sentence, line 6) can be deleted as the word "data" provided earlier already communicates this sufficiently.

Specifically, I'm requesting the authors need to revise that sentence in section 3.1 (page 13, lines 340-341) to state the temporal frequency with which the model data is being used to test the model (being representative of dataset to be provided from geostationary satellite).

And proposing they also make that edit to the new merged sentence on lines 5-6 of the Abstract: -- the suggested edit to insert "2D high temporal resolution" before "column mass loading data".

2) The phrase "nature run" is used throughout the paper, a term I was not familiar with.

I would have expected the terminology "synthetic observations" (or similar term) to convey the fact these are proxy for measurement data, whereas the authors provide a term that refers back to the model run that produced them.

Since none of the 3 expert reviewers have queried this term, I'm not proposing to change that, it obviously not critical to the presentation of the methodology, which particular name is given for a specific aspect.

However, I noticed this term "nature run" is used twice on Page 11 without introduction (section 2.2, lines 290 and 291), whereas it's use later in the manuscript (section 3.1, page 13, line 341) does have a brief introductory explanation of the term.

Please provide an initial definition for that term, and consider whether to change to using "synthetic observations" rather than "nature run", reserving the latter term for where the text is specifically referring to the simulation the generated it.

3) Further edit to improve the revised caption to Figure 2 (page 13)

Reviewer 3 requested to simplify this caption (first bullet point of their comments), and I think they were referring to the "dependence on the assimilation window" which seemed

not to be relevant to that Figure. However, the new shortened title, could be improved to better communicate the relevance of the emission profile shown in the Figure.

The main part of the new revised Figure 2 caption currently says "Hovmoeller plot of the nature run emissions profile used in this study".

Related to comment 2), if the authors prefer to keep to the "nature run" terminology, I'd suggest this Figure caption could be an opportunity to re-iterate that this run is actually providing a dataset to be representative of measurements provided from geostationary satellite – possibly with a descriptor mentioning the specific instrument cited in the Introduction (i.e. "SEVIRI-like" or similar).

Also, even if the "nature run" terminology is retained, including that term prior to "emission profile" is poor grammar, the object of the sentence being that "emission profile".

Specifically I'm suggesting to move "nature run" to instead be after "used in" and "before this study", inserting "the" before it, and "providing the SEVIRI-like synthetic observations for" after it.

Suggest also maybe change "study" to "ESIAS system tests" at the end of the sentence

I mean to revise that first sentence of the Abstract from:

"Hovmoeller plot of the nature run emissions profile used in this study".

Instead to

"Hovmoeller plot of the emissions profile used in the nature run providing the SEVIRI-like synthetic observations for the ESIAS system tests".

or similar wording that will help the time-pressed reader scan the paper to understand the unpacking of the "nature run" term.

4) Page 11, section 2.2, lines 291-294 --- Text here not clear re: explaining the pcc cases.

The word "disjoint" is used in lines 291-292 in reference to a correlation coefficient of zero, but that term seems somehow inappropriate for this case of two uncorrelated datasets.

Furthermore, the follow-on sentence re: how to interpret the pcc < 1 case seems incorrect, unless I misunderstanding the explanation here in relation to the experiment described.

The current text says "indicates that the analysis contains volcanic ash either in model layaers or at times, where no volcanic ash is emitted in the nature run"

That doesn't make sense at all to me – Figure 2 shows the emission profile for the "nature run" and clearly the majority of the re-constraints will be after volcanic ash has been emitted, as the plume is dispersing and the geostationary satellite will be able to identify

where the cloud is dispersing to.

I understand that applying a weighting based on comparison to a test dataset can then weight the predictions towards an optimised input for the predictions.  But this pattern correlation coefficient text here seemed to be describing something else here.

Please re-word these 2 sentences to better communicate the interpretation of these pattern correlation coefficient metrics.

---

## Author Response (AR2)

Reply to the editors final comments.

Topical Editor Decision for "Vertically-resolved probabilistic volcanic ash analysis using the chemical part of the Ensemble for Stochastic Integration of Atmospheric Simulations (ESIAS-chem) version 1.0" by P. Franke, A. C.. Lange and H. Elbern.

I completed two initial Topical Editor reviews of this manuscript (beginning and end of April 2021), a version with revised Abstract and Introduction proceeding then to expert peer review in May 2021.

Three reviewers posted comments during the review period (May to July), with the authors uploading a revised manuscript, ATC version and replies to the reviewers comments at the end of September.

I can see that the authors have replied comprehensively to each of the reviewers' comments, in each case their comments minor in nature, and the manuscript has improved substantially as a result.

All three expert reviewers recommended publication once the required revisions were made, and I can confirm that the authors have addressed these appropriately, both in their replies and in the revised manuscript.

Upon reviewing the revised manuscript however, I did notice two areas where I still feel some aspects are not sufficiently explained.

I am therefore recommending publication once 4 further issues are remedied, or explained, this additional Topical Editor review only for these few minor comments however.

We thank the editor for his final remarks and the thorough revision of the manuscript. We have answered the editor's comments below in blue:

1) On page 13 (lines 340-341), the authors explain:

"Column mass loading of volcanic ash in $[gm^{-2}]$ is extracted as fictional observation data $y_i$ , every 6 hours, from a 'nature run', simulated by the forward model of EURAD-IM. "

I understand the basis for this approach, in using the higher resolution EURAD-IM dispersion model, for two alternative scenarios of a sub-Plinian Eyjafjallajokull eruption, to provide synthetic observations that can then be used to test the ESIAS-chem simulations with the the data assimilation.

What I don't understand though, is that the text states (line 340) that these "fictional observation data" are extracted from the model only every 6 hours, this being presented then as a proxy for a data stream representative of the SEVIRI geostationary satellite.

Clearly the geostationary satellite will provide measurements at a much higher temporal resolution than "every 6 hours", with data every 30 mins or 1 hour being how the system will then be able to adjust/weight its ensemble predictions with the technique described.

I'm assuming that "every 6 hours" must be the authors referring to the data-flows, with perhaps a block of 12 sets of 30-min resolution data extracted every 6 hours. That might well have been obvious to the expert reviewers, but as currently worded, that's a confusing mis-match to the approach for the synthetic observations representative of a geostationary satellite's monitoring of the volcanic plume/cloud's progressing dispersion.

Related to this point, reading the reviewer's reply to the 1st of reviewer 1's comments, to revised the Abstract sentence previously beginning "The system validation", now revised to "Thus, the proposed system" (lines 5-6), I'm suggesting also to highlight the benefits of the geostationary datasets, that primarily being the high temporal resolution that can then constrain how the model predicts that the plume/cloud develops in the initial days (e.g. with the ash particles sedimenting alongside any co-emitted sulphur dioxide oxidising to sulphate aerosol, and the evolving wind shear etc.).

I'm not suggesting to add those specifics of the particles involved, but simply to add "2D high temporal resolution" before "column mass loading data" within that new merged sentence of the Abstract. Also, the word "imagery" (at the end of that sentence, line 6) can be deleted as the word "data" provided earlier already communicates this sufficiently.

Specifically, I'm requesting the authors need to revise that sentence in section 3.1 (page 13, lines 340-341) to state the temporal frequency with which the model data is being used to test the model (being representative of dataset to be provided from geostationary satellite).

And proposing they also make that edit to the new merged sentence on lines 5-6 of the Abstract: -- the suggested edit to insert "2D high temporal resolution" before "column mass loading data".

We thank the editor for his remark. We would like to take the chance to explain our choice of observational data more comprehensively. We have extracted observations only every six hours from the nature run. As we have added in the text (line342-343), we are aware that this time step is significantly larger than for current geostationary satellites. However, we would like to emphasize that the fictional observations, which we use for assimilation, are in general very idealized:
1) We take observations from the full domain, including ash-free areas;
2) We omit any effect of clouds that may influence/hinder the retrieval of a geostationary satellite.
With this setting, the volcanic ash cloud is well observed at observation time. Thus, additional observations in between the six hourly time interval provide no substantial information to the assimilation system. In real applications we have a situation, where both volcanic emissions and domain-wide observations are quasi-continuously given, impeding any attribution of time-height ash emissions to later ash column retrievals. In fact, while the restriction of the data flux

to 6 hour intervals in our analysis reduces the information used, it helps to attribute column mass loading observations to older and recent emissions within the chosen resolution. The coarse observational interval is therefore a mere practical and easily alterable approach, not suggesting any abandonment of observational data streams.

We have added the following text at the end of the respective paragraph:
"Quasi-continuous data streams of emissions and observations, as available in real applications, prohibit any attribution of time-height ash emissions to later ash column retrievals for test purposes. In fact, while the restriction of the data flux to 6 hourly time intervals in our test scenarios reduces the information used, it helps to attribute column mass loading observations to older and recent emissions within the chosen resolution. We found a six hours interval for column mass loading data supply practicable. For later operational purposes the use of the full high frequency data supply is readily adaptable."

As kindly suggested by the editor, we have changed the sentence (line 5-6) in the abstract to:
"Thus, the proposed system addresses the special challenge of analyzing the vertical profile of volcanic ash clouds given only 2D high temporal resolution column mass loading data as retrieved by geostationary satellites."

2) The phrase "nature run" is used throughout the paper, a term I was not familiar with.

I would have expected the terminology "synthetic observations" (or similar term) to convey the fact these are proxy for measurement data, whereas the authors provide a term that refers back to the model run that produced them.

Since none of the 3 expert reviewers have queried this term, I'm not proposing to change that, it obviously not critical to the presentation of the methodology, which particular name is given for a specific aspect.

However, I noticed this term "nature run" is used twice on Page 11 without introduction (section 2.2, lines 290 and 291), whereas it's use later in the manuscript (section 3.1, page 13, line 341) does have a brief introductory explanation of the term.

Please provide an initial definition for that term, and consider whether to change to using "synthetic observations" rather than "nature run", reserving the latter term for where the text is specifically referring to the simulation the generated it.

The editor is right. We have used the term "nature run" before providing a definition, which has now been amended. In general, the fictional observations pose only a small fraction of the data that is used to compare the volcanic ash simulated by the analysis ensemble with the volcanic ash of the nature run (which is considered the truth in this experimental setup). Thus, it is more appropriate to use "nature run" instead of "fictional observations". We have added an explanation of this at the beginning of Section 2.2:
"As ESIAS-chem is tested by identical twin experiments (cf. Section 3 for more details), results of the analysis are compared to a "nature run". In this experimental setting, the nature run is

considered to represent the truth. Synthetic observations are simulated by extracting volcanic ash column mass loading data from this nature run. These synthetic observations show only a small fraction of the data that is used to validate the analysis ensemble. Thus, the following test procedures compare volcanic ash simulated by the analysis ensemble with volcanic ash simulated by the nature run rather than only with the extracted observations."

3) Further edit to improve the revised caption to Figure 2 (page 13)

Reviewer 3 requested to simplify this caption (first bullet point of their comments), and I think they were referring to the "dependence on the assimilation window" which seemed not to be relevant to that Figure. However, the new shortened title, could be improved to better communicate the relevance of the emission profile shown in the Figure.

The main part of the new revised Figure 2 caption currently says "Hovmoeller plot of the nature run emissions profile used in this study".

Related to comment 2), if the authors prefer to keep to the "nature run" terminology, I'd suggest this Figure caption could be an opportunity to re-iterate that this run is actually providing a dataset to be representative of measurements provided from geostationary satellite – possibly with a descriptor mentioning the specific instrument cited in the Introduction (i.e. "SEVIRI-like" or similar).

Also, even if the "nature run" terminology is retained, including that term prior to "emission profile" is poor grammar, the object of the sentence being that "emission profile".

Specifically I'm suggesting to move "nature run" to instead be after "used in" and "before this study", inserting "the" before it, and "providing the SEVIRI-like synthetic observations for" after it.

Suggest also maybe change "study" to "ESIAS system tests" at the end of the sentence

I mean to revise that first sentence of the Abstract from:

"Hovmoeller plot of the nature run emissions profile used in this study".

Instead to

"Hovmoeller plot of the emissions profile used in the nature run providing the SEVIRI-like synthetic observations for the ESIAS system tests".

or similar wording that will help the time-pressed reader scan the paper to understand the unpacking of the "nature run" term.

We thank the editor for this helpful comment. We have changed the figure caption as suggested:

"Hovmoeller-like plot of the volcanic ash emission profile used in the nature run to generate the synthetic observations simulating SEVIRI-like column mass loading data for the ESIAS-chem system tests."

4) Page 11, section 2.2, lines 291-294 --- Text here not clear re: explaining the pcc cases.

The word "disjoint" is used in lines 291-292 in reference to a correlation coefficient of zero, but that term seems somehow inappropriate for this case of two uncorrelated datasets.

Furthermore, the follow-on sentence re: how to interpret the pcc < 1 case seems incorrect, unless I misunderstanding the explanation here in relation to the experiment described.

The current text says "indicates that the analysis contains volcanic ash either in model layaers or at times, where no volcanic ash is emitted in the nature run"

That doesn't make sense at all to me – Figure 2 shows the emission profile for the "nature run" and clearly the majority of the re-constraints will be after volcanic ash has been emitted, as the plume is dispersing and the geostationary satellite will be able to identify where the cloud is dispersing to.

I understand that applying a weighting based on comparison to a test dataset can then weight the predictions towards an optimised input for the predictions. But this pattern correlation coefficient text here seemed to be describing something else here.

Please re-word these 2 sentences to better communicate the interpretation of these pattern correlation coefficient metrics.

We apologize that our description of the pcc score is insufficient. We have changed the sentence containing "disjoint" to:
"If ash cloud covers of the nature run and of the ensemble mean match nowhere, the pattern correlation coefficient equals 0."
Further, we have removed the sentence, in which we discuss the case pcc<1, from this paragraph. We used the pcc score to validate the extend of the volcanic ash cloud and not to make any further judgments on the underlying emission patterns. Thus, the specific sentence is of minor relevance for the study, and we removed it to avoid further confusion.